# FlowBench: A Large Scale Benchmark for Flow Simulation over Complex Geometries

**Ronak Tali[1][†], Ali Rabeh[1][†], Cheng-Hau Yang[1][†], Mehdi Shadkhah[1], Samundra Karki[1], Abhisek Upadhyaya[2], Suriya Dhakshinamoorthy[1], Marjan Saadati[1], Soumik Sarkar[1], Adarsh Krishnamurthy[1], Chinmay Hegde[2], Aditya Balu[1], Baskar Ganapathysubramanian[1*]**

{rtali — arabeh — chenghau — mehdish — samundra}@iastate.edu, au2216@nyu.edu
{snarayan — marjansd — soumiks — adarsh}@iastate.edu, chinmay.h@nyu.edu
{baditya — baskarg}@iastate.edu

[1]Iowa State University, Ames, IA 50011, USA
[2]New York University, New York, NY 10012, USA

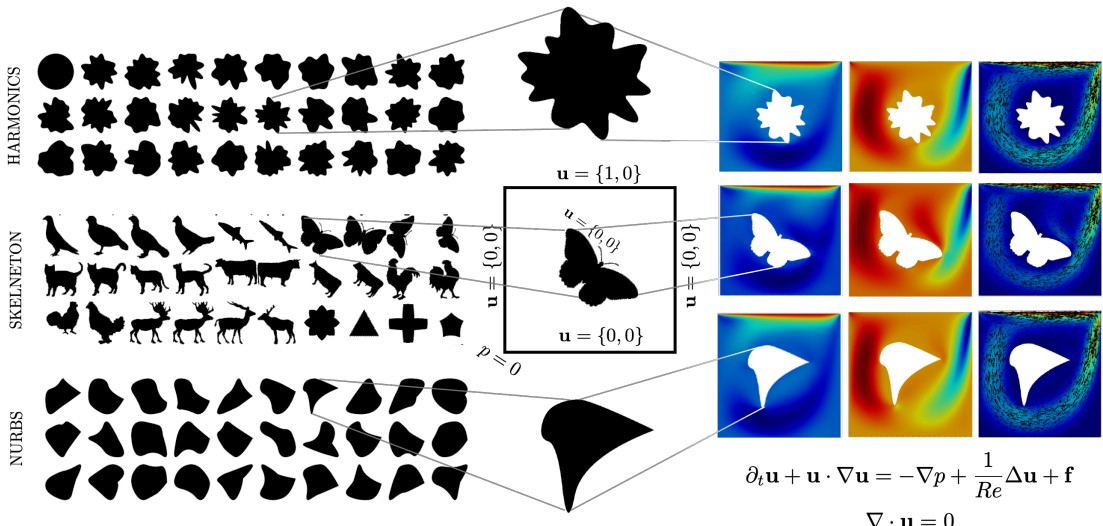

**Figure 1:** *FlowBench offers comprehensive datasets and metrics for assessing neural PDE solvers designed to model flow phenomena around complex objects. It includes three sets of application-relevant geometries with varying complexities and high-fidelity flow simulation data under different forcing conditions. The left panel in the figure above showcases 30 randomly selected shapes from each geometry group. The middle panel provides a close-up of one geometry within the computational domain, highlighting the boundary conditions. The right panel displays the simulation outputs, including velocity results for three samples.*

## Abstract

Simulating fluid flow around arbitrary shapes is key to solving various engineering problems. However, simulating flow physics across complex geometries remains numerically challenging and computationally resource-intensive, particularly when using conventional PDE solvers. Machine learning methods offer attractive opportunities to create fast and adaptable PDE solvers. However, benchmark datasets to measure the performance of such methods are scarce, especially for flow physics across complex geometries. We introduce FlowBench, a dataset for neural simulators with over 10K samples, which is currently larger than any publicly available flow physics dataset. FlowBench contains flow simulation data across complex geometries (*parametric vs. non-parametric*), spanning a range of flow conditions (*Reynolds number and Grashoff number*), capturing a diverse array of flow phenomena (*steady vs. transient; forced vs. free convection*), and for both 2D

---

. [†] These authors contributed equally to this work.

. [*] Corresponding author

and 3D. FlowBench contains over 10K data samples, with each sample the outcome of a fully resolved, direct numerical simulation using a well-validated simulator framework designed for modeling transport phenomena in complex geometries. For each sample, we include velocity, pressure, and temperature field data at 3 different resolutions and several summary statistics features of engineering relevance (such as coefficients of lift and drag, and Nusselt numbers). We envision that FlowBench will enable evaluating the interplay between complex geometry, coupled flow phenomena, and data sufficiency on the performance of current, and future, neural PDE solvers. We enumerate several evaluation metrics to help rank order the performance of current (and future) neural PDE solvers. We benchmark the performance of several methods, including Fourier Neural Operators (FNO), Convolutional Neural Operators (CNO), DeepONets, and recent foundational models. This dataset (here) will be a valuable resource for developing and evaluating AI-for-science approaches, specifically neural PDE solvers, that model complex fluid dynamics around 2D and 3D objects.

## 1 Introduction

Accurate modeling of fluid flow around complicated objects is central to a plethora of applications. In aerospace and automotive applications, flow around wings, and car bodies can significantly impact design and performance Greenblatt and Wygnanski (2000); Spohn and Gilliéron (2002); You and Moin (2008). In civil and environmental engineering applications, understanding fluid flow patterns around structures like buildings (Ramponi and Blocken, 2012) and bridges (Helgedagsrud et al., 2019) is critical from a safety perspective, while flow patterns inside buildings are important from a safety and comfort perspective (Tan et al., 2023). Bio-flow applications, such as flow around fish (Seo and Mittal, 2022; Zhu et al., 2021), birds (Song et al., 2014), insects (Engels et al., 2016), and heart valves (Xu et al., 2021), are all important fields of study. Sports engineering scientists use fluid flow simulations to optimize equipment design, such as bike helmets and golf ball shapes (Ting, 2003). *All these applications involve flow through and across **complex geometries***.

Fluid flow is also often connected with thermal effects. A thermal mismatch between the fluid medium and the object geometry can produce buoyancy-driven phenomena, cause thermal plumes, and impact mixing and thermal transport with significant engineering implications, necessitating a multiphysics approach, where fluid flow and heat transfer are modeled concurrently. For instance, in the semiconductor industry, the design of heat exchangers must account for how the geometry influences both fluid flow and thermal transport (Bhutta et al., 2012; Abeykoon, 2020). Similarly, addressing urban heat island effects requires a deep understanding of the interaction between thermal and flow dynamics in complex urban settings (Priyadarsini et al., 2008; Allegrini and Carmeliet, 2018). Additionally, ensuring indoor comfort and safety, particularly in mitigating the transmission of infectious diseases, depends on accurately modeling the interplay of airflow and temperature within built environments (Bhattacharyya et al., 2020; Tan et al., 2023). *These examples further underscore the necessity of considering multiphyics flow simulations through and across **complex geometries***.

A central challenge in accurately simulating flow patterns in complex geometries is the high resource cost of traditional simulation approaches (Rabeh et al., 2024b). High-fidelity flow (and thermal) simulations often require hours to days (or sometimes even months (Saurabh et al., 2023)) on high-performance computing (HPC) systems. Scientific machine learning (SciML) has emerged as a promising path towards resolving this challenge. By combining training data with domain-specific information (e.g., physical constraints and smoothness assumptions), SciML approaches offer fast simulation, better extrapolation capabilities, and lower data requirements. This includes impactful applications like weather prediction (Rasp et al., 2020) and canonical flows in simple geometries (Khara et al., 2024; Bonnet et al., 2022a; Luo et al., 2024; Xu et al., 2023; Janny et al., 2023; Bonnet et al., 2022b).

Despite these successes, however, there is a notable lack of datasets for flow (and flow-thermal) interactions with complicated geometries. While databases exist for flow past simple shapes such as cylinders (Luo et al., 2024; Xu et al., 2023) and airfoils (Bonnet et al., 2022a,b), there remains a significant gap in datasets involving more diverse and complex shapes. We note the availability of a few datasets that include other geometries, but these geometries are limited to drone shapes (Janny et al., 2023), and canonical car shapes (Ashton et al., 2024b).

In addition to the need for more complex geometries in existing flow datasets, there is also a shortage of multiphysics datasets. By multiphysics, we mean applications that are modeled as a set of (tightly) coupled partial differential equations (PDE), with each PDE modeling a specific physical phenomenon – for instance, Navier-Stokes that models flow phenomena and the advective heat equation that models thermal phenomena. This gap exists because coupling different types of PDEs is inherently challenging and computationally expensive. Solving multiphysics problems requires sophisticated numerical methods and substantial computational resources to accurately simulate each subproblem and capture the interactions between various physical phenomena.

FlowBench seeks to enable the ML community to build the next generation of SciML neural PDE solvers by filling these gaps – complex geometries and multiphysics phenomena. In particular, FlowBench offers:

- **Flow across complex geometries**: We simulate flow and thermal-flow phenomena across a wide range of complex shapes – both parametric and non-parametric – in 2D and 3D. These include simple shapes like ellipses, more complex blobs, and geometries like insects, animals, and birds. Flow across this spectrum of complex objects exhibits a rich array of vortex formation, flow separation, and a range of lift and drag profiles. The diverse shapes and the flow interactions with them provide a rich and intricate dataset for training geometry-aware SciML solvers that can be used for various applications.
- **Multiphysics simulations**: For each shape, we perform a variety of simulations representing flow (i.e., incompressible Navier-Stokes, NS) as well as thermal flow (i.e., coupled Navier-Stokes and Heat Transfer using the Bousinessque coupling, NS-HT) scenarios. Here, we span steady-state and transient behaviors, offering a comprehensive dataset to test and evaluate solvers under various scenarios.
  - For the steady-state case, we consider a variant of the canonical fluid dynamics problem of lid-driven cavity flow (LDC), which is an example of internal flow. The cavity has an object (complex geometry) placed inside (see Figure 8). Additionally, we consider a temperature difference between the cavity walls and the object, producing thermal and flow coupling. For each object, we simulate across a range of Reynolds number $Re \in [10^1, 10^3]$, and Grashof number $Gr \in [10^1, 10^7]$. This range offers a variety of forced $\left(Gr/Re^2 \approx 0.1\right)$, mixed, and free convection $\left(Gr/Re^2 \approx 10\right)$ scenarios. In this case, there are nearly 9000 unique samples for 2D and 500 unique samples for 3D.
  - For the transient case, we consider a variant of another canonical fluid dynamics problem – flow past a bluff object (FPO) – which is an example of external flow. Here, we consider flow moving past a stationary (complex geometry) object that is placed in a large domain. The fluid exhibits various intricate time-dependent patterns as it moves past the object. For each object, we simulate across a range of Reynolds number $Re \in [10^2, 10^3]$, offering an array of vortex-shedding frequencies and other time-dependent patterns. In this case, there are over 1000 unique samples in 2D.

  We provide field data of velocity, pressure, and temperature for each of the over $10K$ simulations. Additionally, we provide summary engineering features, including the coefficient of lift ($C_L$), drag ($C_D$), and the average heat transfer (Nusselt number, $Nu$) from the surface of the complex object. We provide our dataset on huggingface at https://huggingface.co/datasets/BGLab/FlowBench/tree/main as a benchmark for others interested in the development and evaluation of SciML models.
- **Benchmark metrics and comparisons**: Besides the dataset, we also include workflows to train select group of neural operators – Fourier Neural Operators (FNO), Convolutional Neural Operators (CNO), and Deep Operator Networks (DeepONets). Neural operators can be trained using our code for the steady-state case and study the comprehensive evaluation metrics that we recommend in Section 4.6. We also suggest a hierarchy of in-distribution and out-of-distribution tests to evaluate the generalizability of these models.

The dataset (consisting of over $10K$ samples), evaluation metrics, workflows, and trained models together make FlowBench a valuable tool for the ML community to create SciML solvers of coupled phenomena involving complex geometries. All data, visualizations, models, and model evaluations are available on the FlowBench website.

**Table 1:** *Comparison of incompressible flow simulation data in Curated Dataset, Graph-Mesh, CFDBench, AirfRANS, PDEBench, Eagle, MegaFlow, Fluid Flow Dataset, ScalarFlow, WindsorML, AhmedML, DrivAerML, BubbleML, and our FlowBench. An orange check mark indicates a restricted family of shapes (Airfoil: AirRANS and Graph-Mesh; Cylinder: CFDBench, Cylinders and Ellipses: MegaFlow; Drone geometry: Eagle; Windsor bodies: WindsorML; Ahmed bodies: AhmedML; DrivAer bodies: DrivAerML).*

| Name | Dim. | Size | Geometry | # Geometries | Multiphysics |
|---|---|---|---|---|---|
| **Curated Dataset** (McConkey et al., 2021) | 2 | 116 | ✗ | 0 | ✗ |
| **Graph-Mesh** (Bonnet et al., 2022b) | 2 | 230 | ✓ | Few airfoil shapes | ✗ |
| **CFDBench** (Luo et al., 2024) | 2 | 739 | ✓ | 1 | ✗ |
| **AirfRANS** (Bonnet et al., 2022a) | 2 | 1000 | ✓ | 375 airfoil shapes | ✗ |
| **PDEBench** (Takamoto et al., 2023) | 2 | 1000 | ✗ | 0 | ✗ |
| **Eagle** (Janny et al., 2023) | 2 | 1200 | ✓ | 600 drone shapes | ✗ |
| **MegaFlow** (Xu et al., 2023) | 2 | 3000 | ✓ | 3000 ellipses and circles | ✗ |
| **Fluid Flow** (Jakob et al., 2020) | 2 | 8000 | ✗ | 0 | ✗ |
| **ScalarFlow** (Eckert et al., 2019) | 3 | 100 | ✗ | 0 | ✓ |
| **WindsorML** (Ashton et al., 2024a) | 3 | 355 | ✓ | 355 Windsor bodies | ✗ |
| **AhmedML** (Ashton et al., 2024b) | 3 | 500 | ✓ | 500 Ahmed car bodies | ✗ |
| **DrivAerML** (Ashton et al., 2024c) | 3 | 500 | ✓ | 500 DrivAer car bodies | ✗ |
| **BubbleML** (Hassan et al., 2023) | 2,3 | 79 | ✗ | 0 | ✓ |
| **FlowBench** (ours) | 2,3 | **10,000** | ✓ | 300 different shapes | ✓ |

## 2 Related Work

Most publicly available benchmark datasets are summarized and compared in Table 1. PDEBench (Takamoto et al., 2023) presents a dataset consisting of simulations run for a wide range of PDEs, not just flow physics. It includes simulations for both compressible and incompressible Navier-Stokes problems, in both two and three dimensions. CFDBench (Luo et al., 2024) is a fluid flow-focused dataset containing a total of 739 cases. These cases are distributed across various setups: lid-driven cavity (with varying density and viscosity over 25 different length and width combinations), tube flow (varying density, viscosity, and geometry, with 50 different inlet velocity conditions), dam flow, and cylinder flow specialties. Megaflow2D (Xu et al., 2023) is a comprehensive collection of 3000 cases for 2D Navier-Stokes problems. Each case features different geometrical configurations, including circles, ellipses, and nozzles. All simulations are performed at a fixed Reynolds number of 300.

McConkey et al. (2021) focuses exclusively on 2D turbulence simulations, covering 841 different cases across 29 flow scenarios such as a periodic hill, square duct, parametric bump, converging-diverging channel, and curved backward-facing step. Each scenario includes 29 simulations with varying parameters, such as Reynolds numbers, producing 841 data samples. The Graph-Mesh (Bonnet et al., 2022b) dataset simulates incompressible 2D Navier-Stokes problems at high Reynolds numbers (greater than $10^6$) exclusively for Aerofoils. It varies the angle of attack, inlet velocity, Reynolds number, and Mach number for a total of 230 simulations. A distinguishing feature is the calculation of drag and lift after fitting the SciML models, not just the solutions. WeatherBench (Rasp et al., 2020) condenses raw weather data from the ERA5 dataset by reducing resolution levels to fit within a GPU. Along with discrete weather-based variables, WeatherBench runs simulations for a range of $u, v$ velocities, temperature, and vorticity. It uses the entire world as its grid and provides both 2D and 3D data post-processed from the ERA5 dataset.

ScalarFlow (Eckert et al., 2019) is the multiphysics simulation dataset coupling Navier-Stokes with density distribution solving, focusing on buoyancy-driven smoke plume reconstructions. It provides volumetric 3D flow reconstructions for complex buoyancy-driven flows transitioning to turbulence. The Fluid Flow Dataset (Jakob et al., 2020) contains 8000 unsteady 2D fluid flow simulations, each with 1001 time steps, parameterized by Reynolds number. BubbleML (Hassan et al., 2023) is another multiphysics dataset to couple two-phase

Navier-Stokes and energy equations. It includes 79 simulations covering nucleate pool boiling, flow boiling, and sub-cooled boiling under various gravity conditions, flow rates, sub-cooling levels, and wall superheat. EAGLE (Janny et al., 2023) is a large (over 1.1M 2D simulations) dataset that models the airflow generated by 2D UAV moving in a 2D environment with different boundary geometries. This dataset captures the highly turbulent flow consisting of non-periodic eddies induced by the varying geometries.

We identify limitations of these existing datasets that motivate the curation of FlowBench.

- Restricted Geometries: Many datasets are confined to simple or canonical geometries such as airfoils, cylinders, or standard shapes. For instance, AirFRANS focuses exclusively on 375 airfoil shapes, while CFDBench simulates only one geometry type (cylinders). Similarly, datasets like MegaFlow and AhmedML emphasize restricted geometry families like ellipses and Ahmed bodies, respectively. These datasets lack the diversity needed to explore flow dynamics over highly complex or non-parametric geometries, which are common in real-world engineering problems.

- Limited Multiphysics Coverage: While datasets like ScalarFlow and BubbleML include multiphysics simulations, their scope is narrow, focusing only on specific phenomena like buoyancy-driven flows or two-phase boiling. Most other datasets, including PDEBench and Eagle, completely omit the coupling of flow with thermal or other physical phenomena, which are critical in areas such as thermal management and environmental modeling.

- Small Dataset Size: The scale of existing datasets often limits their utility for training data-intensive machine learning models. For example, CFDBench includes only 739 cases, and Graph-Mesh contains 230 cases, both of which are inadequate for benchmarking modern neural PDE solvers that rely on large-scale data to achieve generalizability and robustness.

- Lack of Diverse Flow Regimes: Many datasets focus on specific flow regimes (e.g., laminar or turbulent) and neglect transitional flows or mixed convection scenarios. For instance, MegaFlow provides simulations at a fixed Reynolds number, missing the opportunity to examine the impact of varying flow conditions on solver performance.

- Resolution and Usability: While some datasets include high-resolution simulations, they often lack multi-resolution data that can facilitate super-resolution studies or efficiency testing. Additionally, the absence of consistent and reusable formats for dataset access hampers usability in machine learning workflows.

## 3 Mathematical Formulation of Simulation Approach for Creating FlowBench

Simulating incompressible flows over complex geometries is challenging, especially with traditional body-fitted mesh methods. These approaches often struggle to handle intricate shapes, as creating a mesh that accurately conforms to irregular geometries requires significant manual adjustment to ensure stability and convergence, making large-scale simulations labor-intensive and costly (Verzicco, 2023; Mittal and Seo, 2023; George et al., 1991; Owen, 1998; Persson, 2005; Boelens et al., 2009; Kurtulus, 2015). Although advances in automated mesh generation (George et al., 1991; Simonovski and Cizelj, 2011; Beaufort et al., 2020) have eased some difficulties, achieving high-quality meshes for complex structures without human intervention remains elusive.

A promising alternative is the Immersed Boundary Method (IBM), first developed by Peskin (Peskin, 1972), which enables simulations across complex geometries by embedding the shape within a fixed mesh. The Shifted Boundary Method (SBM), a robust variant of IBM, further improves this approach by imposing boundary conditions on a surrogate rather than the actual boundary. In SBM, the surrogate boundary aligns with a Cartesian mesh, and boundary conditions are adjusted using Taylor expansions, effectively transforming the problem to resemble a body-fitted domain while streamlining mesh generation (Main and Scovazzi, 2018b,a; Atallah et al., 2020, 2021; Yang et al., 2024a). This surrogate is carefully chosen to minimize numerical errors, ensuring accuracy and stability. Particularly effective when applied with adaptive octree meshes, SBM allows local refinement near complex geometries, reduces computational demands, and enhances scalability, making it ideal for complex fluid flow and multiphysics simulations across varied Reynolds numbers and configurations.

For completeness, we briefly detail the Finite Element approach – specifically, the additional terms needed in Shifted Boundary Method over the standard stabilized finite element formulation – used to create the dataset. In the formulations that follow, $w_i$ is defined as the test function for the $i$-th component of the momentum equation, where $i$ runs from 0 to $\text{DIM} - 1$ (with DIM representing the spatial dimension), corresponding to the $x$-, $y$-, and $z$-directions. Similarly, $u_i$ denotes the corresponding velocity components. These conventions are standard in the FEM literature. We refer the interested reader to (Yang et al., 2024b) for a detailed discussion of the approach, implementation details, and exhaustive validation of the accuracy of the SBM approach.

## 3.1 Formulation of SBM for Navier Stokes

In SBM, additional terms are incorporated into the standard stabilized finite element formulations for solving the Navier-Stokes equations (see Section 2 in (Yang et al., 2024b)). These terms include a *consistency* term (which appears due to integration by parts operation), an *adjoint consistency* term (which is included to ensure optimal convergence rates), and the *penalty* term (which ensures that as the mesh size is reduced, the boundary conditions asymptote to the true boundary conditions) :

$$
\widetilde{B_{NS}^{VMS}} = B_{NS}^{VMS} \underbrace{- \left\langle w_i^{c,h}, \frac{1}{Re}\left(\frac{\partial u_i^{c,h}}{\partial x_j} + \frac{\partial u_j^{c,h}}{\partial x_i}\right)\tilde{n}_j - p^{c,h}\tilde{n}_i \right\rangle_{\tilde{\Gamma}_{D,h}}}_{Consistency\ term}
$$

$$
\underbrace{- \left\langle \frac{1}{Re}\left(\frac{\partial w_i^{c,h}}{\partial x_j} + \frac{\partial w_j^{c,h}}{\partial x_i}\right)\tilde{n}_j + q^{c,h}\tilde{n}_i, u_i^{c,h} + \frac{\partial u_i^{c,h}}{\partial x_j}d_j \right\rangle_{\tilde{\Gamma}_{D,h}}}_{Adjoint\ consistency\ term}
$$

$$
\underbrace{+ \frac{\beta}{h \cdot Re}\left\langle w_i^{c,h} + \frac{\partial w_i^{c,h}}{\partial x_j}d_j, u_i^{c,h} + \frac{\partial u_i^{c,h}}{\partial x_j}d_j \right\rangle_{\tilde{\Gamma}_{D,h}}}_{Penalty\ term}, \tag{1}
$$

and

$$
\widetilde{F_{NS}^{VMS}} = F_{NS}^{VMS} \underbrace{- \left\langle \frac{1}{Re}\left(\frac{\partial w_i^{c,h}}{\partial x_j} + \frac{\partial w_j^{c,h}}{\partial x_i}\right)\tilde{n}_j + q^{c,h}\tilde{n}_i, g_i \right\rangle_{\tilde{\Gamma}_{D,h}}}_{Adjoint\ Consistency\ Term}
$$

$$
\underbrace{+ \frac{\beta}{h \cdot Re}\left\langle w_i^{c,h} + \frac{\partial w_i^{c,h}}{\partial x_j}d_j, g_i \right\rangle_{\tilde{\Gamma}_{D,h}}}_{Penalty\ Term}, \tag{2}
$$

where $g_i$ is the boundary condition we want to apply, $\tilde{n}_i$ is the unite outward-pointing normal, $h$ is the element size, $\beta$ is the penalty parameter for the Navier-Stokes equation, $B_{NS}^{VMS}$ is the bilinear weak form for Navier-Stokes without SBM, and $F_{NS}^{VMS}$ is the linear weak for Navier-Stokes form without SBM. The formulation for Navier-Stokes with SBM can be expressed as:

$$
\widetilde{B_{NS}^{VMS}} - \widetilde{F_{NS}^{VMS}} = 0. \tag{3}
$$

Readers interested in the derivation of the consistency term, adjoint-consistency term, and penalty term for weakly enforcing shifted boundary conditions are referred to Section A.1.

## 3.2 Formulation of SBM for Heat Transfer

Similar to the Navier-Stokes equations, we use SBM to apply Dirichlet boundary conditions in the energy equation. Essentially, we use SBM to set the temperature to a desired value ($T_D$) on the geometries. The

formulation for energy equation with SBM is:

$$\widetilde{B_{HT}^{VMS}} - \widetilde{F_{HT}^{VMS}} = 0, \tag{4}$$

with

$$\widetilde{B_{HT}^{VMS}} = B_{HT}^{VMS} - \underbrace{\frac{1}{Pe}\left\langle l^{c,h}, \frac{\partial T^{c,h}}{\partial x_j}\tilde{n}_j \right\rangle_{\tilde{\Gamma}_{D,h}}}_{\text{Consistency term}} - \underbrace{\frac{1}{Pe}\left\langle \frac{\partial l^{c,h}}{\partial x_j}\tilde{n}_j, T^{c,h} + \frac{\partial T^{c,h}}{\partial x_j}d_j \right\rangle_{\tilde{\Gamma}_{D,h}}}_{\text{Adjoint consistency term}}$$
$$+ \underbrace{\frac{\alpha}{h \cdot Pe}\left\langle l^{c,h} + \frac{\partial l^{c,h}}{\partial x_j}d_j, T^{c,h} + \frac{\partial T^{c,h}}{\partial x_j}d_j \right\rangle_{\tilde{\Gamma}_{D,h}}}_{\text{Penalty term}}, \tag{5}$$

and

$$\widetilde{F_{HT}^{VMS}} = F_{HT}^{VMS} - \underbrace{\frac{1}{Pe}\left\langle \frac{\partial l^{c,h}}{\partial x_j}\tilde{n}_j, T_D \right\rangle_{\tilde{\Gamma}_{D,h}}}_{\text{Adjoint consistency term}} + \underbrace{\frac{\alpha}{h \cdot Pe}\left\langle l^{c,h} + \frac{\partial l^{c,h}}{\partial x_j}d_j, T_D \right\rangle_{\tilde{\Gamma}_{D,h}}}_{\text{Penalty term}}, \tag{6}$$

where $T_D$ is the boundary condition we want to apply, $\alpha$ is the penalty parameter for energy equation, $Pe$ is Peclet number, which is $0.7Re$ inside our simulations, $B_{HT}^{VMS}$ is the bilinear weak form for Heat Transfer without SBM, and $F_{HT}^{VMS}$ is the linear weak form for Heat Transfer without SBM.

## 4 FlowBench

### 4.1 Geometries

Our dataset includes three distinct categories of geometries, namely **G1**, **G2**, and **G3** as illustrated in Figure 2. The first set of geometries, **G1**, consists of parametric shapes generated using Non-Uniform Rational B-Splines (NURBS) curves. NURBS are mathematical representations used in computer graphics and CAD systems to generate and represent curves and surfaces. They offer great flexibility and precision in modeling complex shapes. Each NURBS curve is defined by a set of control points, the degree of the basis function, and knot vectors (Piegl and Tiller, 2012). We use a uniform knot vector with a second-order (quadratic) basis function, which remains fixed. However, the positions of eight control points are randomly varied to produce a variety of curves. We ensure that the shapes are smooth and do not have any discontinuities or self-intersections. All shapes are normalized to fit within the unit hypercube, $[0, 1]^2$. We provide the shapes as well as code for recreating these geometries.

The next set of geometries, **G2**, consists of parametric shapes generated using spherical harmonics (Wei et al., 2018). We randomly select $N = 8, ..., 15$ harmonics with amplitudes $(a_n, b_n)$ ranging from 0 to 0.2. The radial function $r(t) = 0.5 + \sum_{n=1}^{N} (a_n \cos(nt) + b_n \sin(nt))$ then defines the shape; and is computed at 500 evenly spaced points in $t \in [0, 2\pi]$. We normalize $r(t)$ so that any surface point is within a distance of 0.5 from the center of the shape, $r(t) = 0.5 \left( \frac{r(t)}{r_{\max}} \right)$. We provide shapes as well as code for recreating these geometries.

The last set of geometries, **G3**, consist of non-parametric shapes sampled from the grayscale dataset in SkelNetOn (Demir et al., 2019; Atienza, 2019). We apply a Gaussian blur filter with a scale of 2 to smoothen out some of the thin features of the object. This choice of $\sigma = 2$ was empirically determined to balance the smoothing of small-scale, potentially noisy features—which, if left unfiltered, could lead to jagged or irregular contours during contour extraction and meshing—against the need to preserve the primary geometric characteristics of the shapes. This ensures the shape remains consistent across the three resolutions we provide data for. We then scale the shape to ensure it is contained in the unit hypercube $[0, 1]^2$.

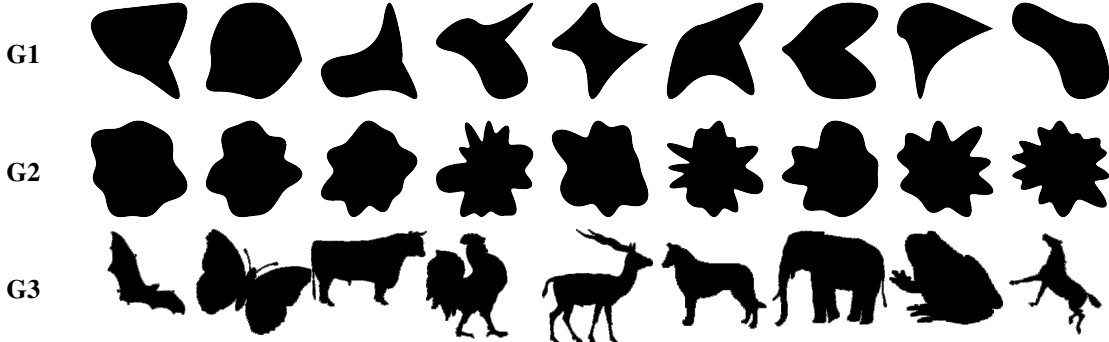

**Figure 2:** *Examples of the diverse and complex geometries in FlowBench using 9 samples from each of the three groups. The first row corresponds to geometries from the nurbs group G1, the second row to the spherical harmonics group G2, and the third row to the skelneton group G3.*

## 4.2 Simulation Framework and Compute effort

Our simulation framework is a highly parallel octree-based CFD and multiphysics code. We account for complex geometries using a robust variant of the immersed boundary method, called the Shifted Boundary Method (SBM) (Main and Scovazzi, 2018b,a). For readers interested in our framework's SBM formulation and implementation, please refer to (Yang et al., 2024a,c). Our octree-based framework has been extensively validated and used in various applications like industrial-scale CFD simulations (Saurabh et al., 2021a), two-phase flow (Khanwale et al., 2023), buoyancy-driven flows (Xu et al., 2019), and coupled multiphysics application (Kim et al., 2024). For additional validation of the incompressible flow and thermal incompressible flow solving using SBM, please refer to Section 4.3 in the appendix. All the 2D LDC cases are simulated using a uniform mesh of $512 \times 512$. All the 2D FPO cases are simulated using an adaptive mesh that resolved to the Kolmogorov length scale close to the object. All the 3D LDC cases are simulated using an adaptive mesh that resolved to $2\times$ the Kolmogorov scale. Illustrations of the mesh are provided in Section A.2. We deployed this framework on one of the largest academic supercomputing clusters in the US, called TACC Frontera (Stanzione et al., 2020). These simulations required about 65K node hours of runtime.

## 4.3 Validation of the numerical framework used to create the dataset

We rigorously validate our SBM framework using results from existing literature. See (Yang et al., 2024c) where we have an exhaustive set of validations. In particular, for cases lacking validation within the literature, such as complex geometry simulations, we compare our SBM results with those obtained from the Boundary-Fitted Method (BFM). This comparison is especially valuable as the BFM requires a tedious and time-consuming process of manually creating body-fitted meshes to capture the geometry accurately. Crafting these meshes is often challenging and labor-intensive, demanding extensive domain knowledge and iterative refinement to ensure mesh quality and alignment with complex boundaries. For completeness, we showcase a representative set of validation examples of the SBM approach.

**LDC (Lid-driven cavity flow), NS:** We simulate a disk with a diameter $D = \frac{L}{3}$ placed at the center of the lid-driven cavity, where $L$ is the chamber's box length. Our CFD results match against detailed simulations available in literature (Huang and Lim, 2020) for this canonical problem, as shown in Figure 3.

**LDC (Lid-driven cavity flow), NSHT:** To validate our multiphysics simulation framework, we select a case from (Chen et al., 2020) to compare with. Here, a heated circle is placed at the center of the chamber, with a radius of 0.2L, where L represents the length of the chamber. We evaluate the local Nusselt number on the bottom wall and report an excellent match with (Chen et al., 2020) in Figure 4 at two distinct operating conditions.

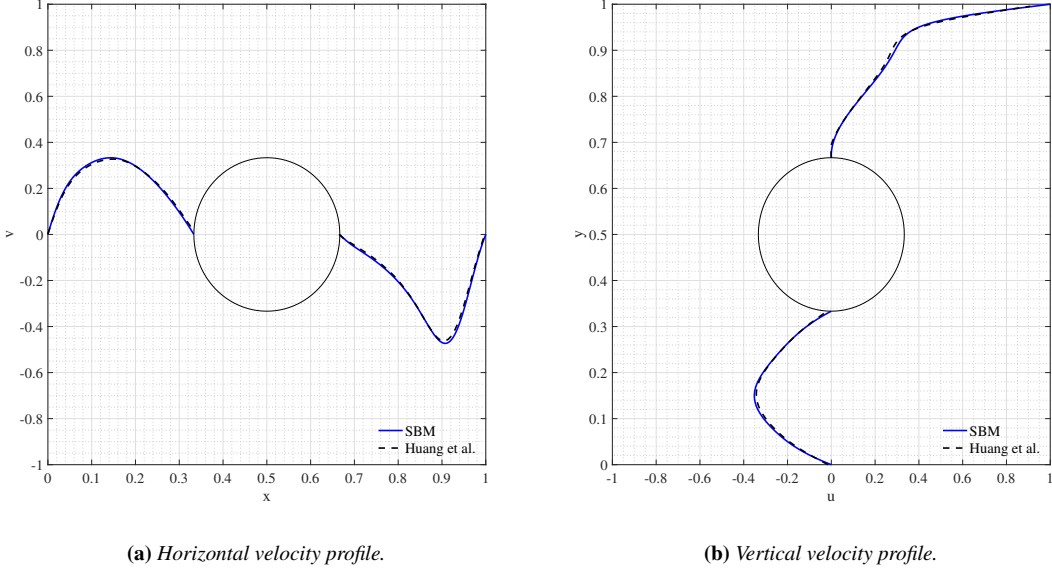

**(a)** *Horizontal velocity profile.*

**(b)** *Vertical velocity profile.*

**Figure 3:** *Comparison of velocity profiles for lid-driven cavity flow with a circular disk at Re = 1000 against results from the literature.*

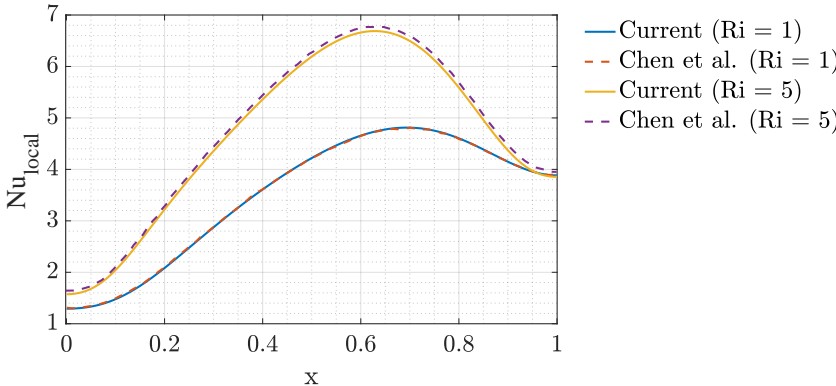

**Figure 4:** *Comparison of the Nusselt number profile along the bottom wall of the domain. The simulation results are validated against data from (Chen et al., 2020) to demonstrate the accuracy of the current multiphyics (NSHT) model.*

**FPO (Flow past object):** For the flow past (bluff) geometries, we tested several shapes. We first compared against the canonical case of flow past a circular disk and evaluated at two different Reynolds numbers: 100 and 1000. Our drag coefficients and Strouhal numbers matched well with the literature, as shown in Table 2 and Table 3.

**Complex Geometries:** To evaluate our framework's performance with complex geometries, we first present further validation of the Shifted Boundary Method (SBM) by comparing it with the Boundary-Fitted Method (BFM) in the context of lid-driven cavity simulations shown in Figure 5. For the boundary-fitted mesh, we used a mesh size of approximately $\frac{1}{2^9}$, matching the element size employed in the SBM simulations.

Additionally, we validated the solver's performance for complex geometries by comparing our results to previous research. Specifically, we simulated flow past a D-shaped cylinder at various Reynolds numbers and compared the results with those from (Shao et al., 2020), as shown in Table 4. The flow visualizations are

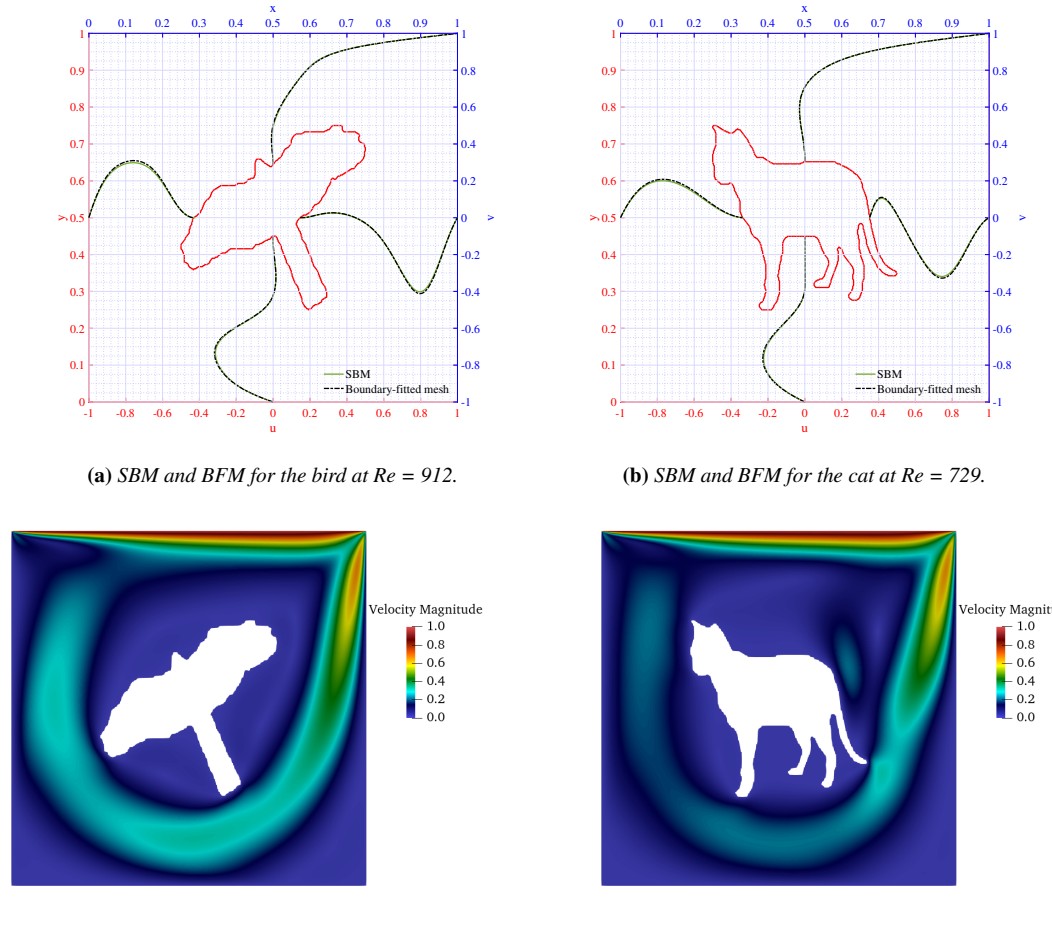

**(a)** *SBM and BFM for the bird at Re = 912.*

**(b)** *SBM and BFM for the cat at Re = 729.*

**(c)** *Flow visualization for the bird at Re = 912.*

**(d)** *Flow visualization for the cat at Re = 729.*

**Figure 5:** *Comparison between results from BFM and SBM in lid-driven cavity simulations for the bird and cat cases.*

**Table 2:** *Comparison of drag coefficients and strouhal numbers for flow past a 2D cylinder at Re = 100.*

|  | **Cd** | **St** |
| --- | --- | --- |
| Current | 1.35 | 0.167 |
| Mittal *et al.* (Mittal et al., 2008) | 1.35 | 0.165 |
| Henderson *et al.* (Henderson, 1995) | 1.35 | - |
| Luo *et al.* (Luo et al., 2009) | 1.35 | 0.159 |
| Kamensky *et al.* (Kamensky et al., 2015) | 1.386 | 0.170 |
| Main *et al.* (Main and Scovazzi, 2018a) | 1.36 | 0.169 |
| Kang *et al.* (Kang and Masud, 2021) | 1.374 | 0.168 |

illustrated in Figure 6. Our findings closely align with the literature, demonstrating that an increase in Reynolds number corresponds to a higher Strouhal number. Readers interested in further validating our simulation framework for complex geometries are encouraged to read (Yang et al., 2024c) and visit our website.

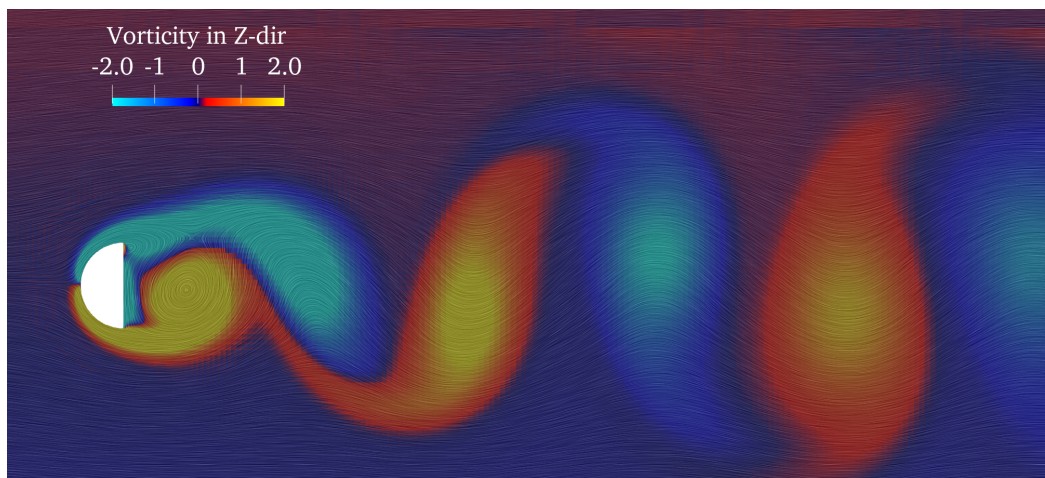

**Figure 6:** *Flow visualization for a D-shaped cylinder at Re = 100.*

**Table 3:** *Comparison of drag coefficients and strouhal numbers for flow past a 2D cylinder at Re = 1000.*

|  | Cd | St |
|---|---|---|
| Current | 1.52 | 0.239 |
| Mittal *et al.* (Mittal et al., 2008) | 1.45 | 0.230 |
| Henderson *et al.* (Henderson, 1995) | 1.51 | - |
| Luo *et al.* (Luo et al., 2009) | 1.56 | 0.235 |
| Cheny *et al.* (Cheny and Botella, 2010) | 1.61 | 0.251 |
| Jester *et al.* (Jester and Kallinderis, 2003) | 1.51 | 0.25 |

**Table 4:** *Strouhal numbers for flow past a D-shaped cylinder*

| Re | Current | (Shao et al., 2020) |
|---|---|---|
| 46 | 0.1363 | 0.1316 |
| 100 | 0.2006 | 0.1910 |
| 160 | 0.2112 | 0.2042 |

### 4.4 Flow Physics Problems: Domain, Boundary Conditions, and Outputs

Our FlowBench dataset presents a comprehensive suite of simulations across 300 complex geometries, capturing a diverse range of physical phenomena, including Navier-Stokes and thermally coupled flows. This dataset addresses a spectrum of canonical problems—such as flow past obstacles and lid-driven cavity flow with internal objects—spanning both two-dimensional and three-dimensional domains. The dataset accommodates various flow regimes, encompassing forced and natural convection as well as transitions from laminar to turbulent flows. A structured, hierarchical overview of the dataset is provided in Figure 7, illustrating its extensive applicability in benchmarking complex fluid dynamics and heat transfer scenarios.

In the 2D LDC setup, a square features three stationary walls and one moving lid, with a domain size of [0, 2] × [0, 2]. An object is placed in the middle of the flow within the chamber. Examples of LDC simulations showing streamlines and y-direction velocities, along with the drag coefficient ($C_D$) and lift coefficient ($C_L$) values, are shown in Figure 8. Increasing the Reynolds number brings the vortices closer to the right wall, with additional vortices forming at the bottom-left and bottom-right corners. With increasing Reynolds numbers, the smaller viscous forces acting on the geometries decrease both $C_D$ and $C_L$.

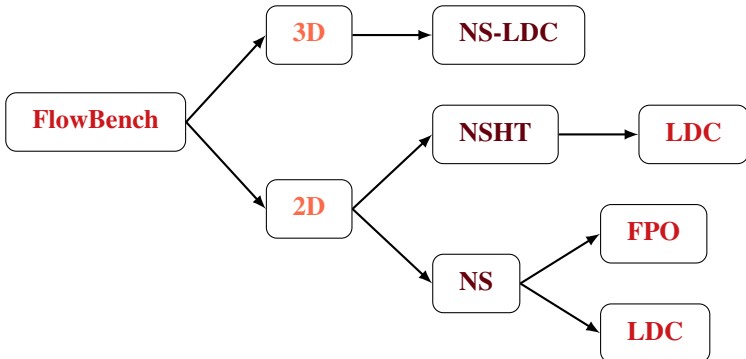

**Figure 7:** *Hierarchical structure of the FlowBench framework. The diagram illustrates the categorization into 2D and 3D simulations, with further subdivisions based on Navier-Stokes (NS) and Navier-Stokes coupled with Heat Transfer (NSHT) under different test cases, such as the Lid-Driven Cavity (LDC) and Flow Past Object (FPO). Each category represents a specific configuration for benchmarking complex fluid dynamics and heat transfer simulations.*

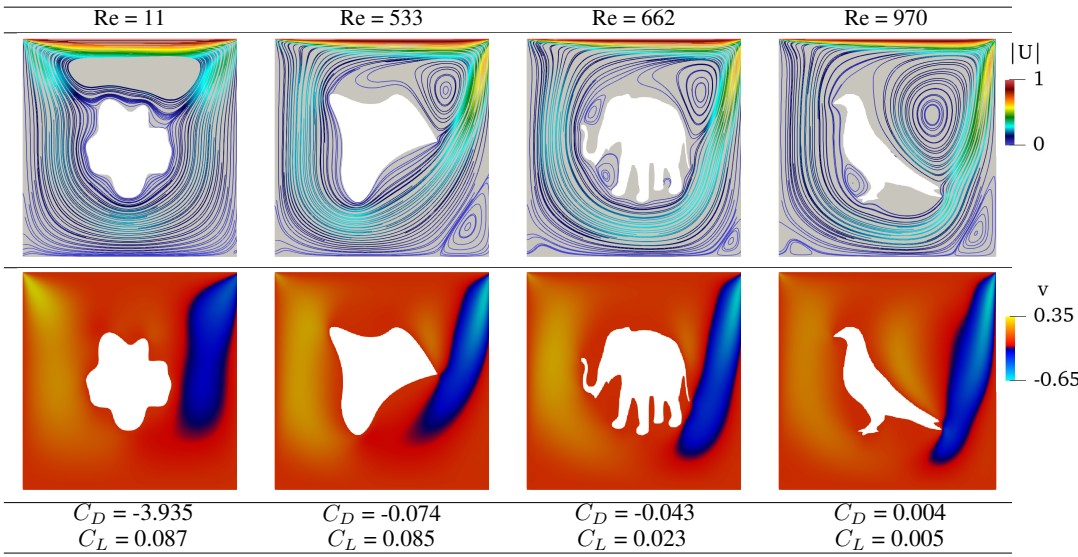

**Figure 8:** *Drag coefficients ($C_D$), and lift coefficients ($C_L$) for different shapes and different Reynolds numbers in pure-NS LDC simulations.*

For the 2D Navier-Stokes and heat transfer (NSHT) lid-driven cavity (LDC) problem, the bottom wall is set to a temperature of 1, while the top wall is set to 0. The left and right walls have zero-flux temperature boundary conditions. The object surface is set to a temperature of 0. This setup produces a combination of forced convection (flow due to the moving lid) and Rayleigh-Benard instabilities (flow due to buoyancy-driven natural convection). Examples of NSHT-LDC simulations with fixed Reynolds number, showing streamlines, temperatures, and the values of $C_D$, $C_L$, and Nu, are presented in Figure 9. We use a non-dimensional number, Richardson number, defined as $Ri = Gr/Re^2$, representing the ratio between buoyancy and inertial force. As $Ri$ increases, we observe higher values of $C_D$ and $C_L$. A higher Richardson number indicates that buoyancy effects are more significant than the forced flow. The heated fluid rises more strongly, creating greater circulation within the chamber. The increased circulation results in stronger forces acting on the object, leading to higher $C_D$ and $C_L$.

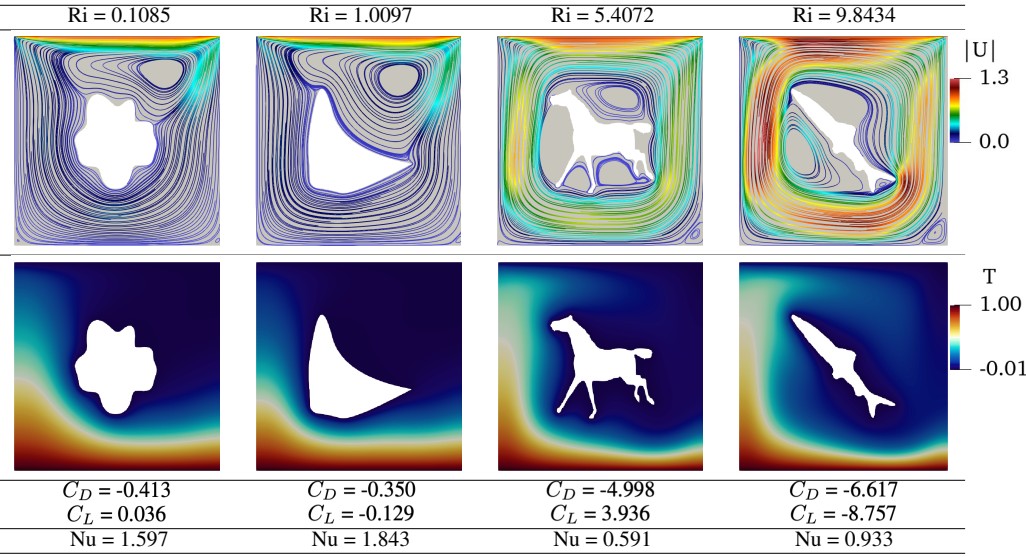

**Figure 9:** *Drag coefficients ($C_D$), lift coefficients ($C_L$), and Nusselt numbers (Nu) for different shapes and different Richardson numbers in NSHT LDC simulations (fixed Re = 100).*

Apart from the NSHT-LDC simulations with a fixed Reynolds number, we also conduct simulations with randomly varying Reynolds and Richardson numbers. We pick up one geometry to demonstrate in Figure 10. This showcase examines four cases with progressively increasing Reynolds and Richardson numbers. In Case 1, the high-speed flow predominantly occurs at the top of the geometry, resulting in a negative $C_D$ due to the flow applying force on the upper part of the geometry. As we increase the Reynolds number in Case 2, the flow shifts towards the bottom part of the geometry, applying force in the positive y-direction, leading to a change in the sign of $C_D$ from Case 1 to Case 2. The same as pure-NS flow, we observe a high-speed jet in the y-direction near the left-hand side wall due to the higher Reynolds number in Case 2. The magnitudes of $C_D$ and $C_L$ decrease from Case 1 to Case 2 because the viscous forces are reduced as the Reynolds number increases. Given that the Reynolds numbers in Case 2 and Case 3 are close, the reduction in force due to the increasing Reynolds number is mitigated, and the Richardson number plays a crucial role in enhancing flow circulation, leading to an increase in $C_D$ and $C_L$. However, from Case 3 to Case 4, the drag force decreases due to the higher Reynolds number, as observed in Figure 8. Additionally, as the Richardson number increases, the temperature convect to higher positions, progressing from Case 1 to Case 4.

For the FPO setup, we consider a domain $[0, 64] \times [0, 16]$ with the object placed at $(6, 8)$. A parabolic velocity inlet boundary condition is applied on the left, no-slip boundary conditions are applied on the top and bottom walls, and a zero pressure boundary condition is applied on the right. The large domain size for the FPO problem is chosen to capture as much physical detail as possible in our dataset. We report a smaller cropped-out region of size $[0, 16] \times [0, 4]$ from this dataset, representing a tradeoff between dataset size and the amount of physics captured. Figure 11 shows time snapshots of representative shapes showing vortex shedding, as the flow rotates and stretches around the object. Movies of the time-dependent flow can be seen on the FlowBench website.

For the simulations we performed in 2D, we ensure the fluid mesh's resolution is fine enough for the Direct Numerical Simulation (DNS) simulations. DNS is an expensive high-fidelity computational approach that solves the full Navier-Stokes equations, resolving all scales of motion in a fluid without additional turbulence modeling. The Kolmogorov scale is crucial in turbulence theory as it represents the smallest length scale that must be resolved in DNS to capture all aspects of turbulence accurately. This scale can be expressed in terms of the Reynolds number as $\eta \sim {}^L/_{Re^{3/4}}$, where $L$ is the characteristic large-eddy length scale. In our

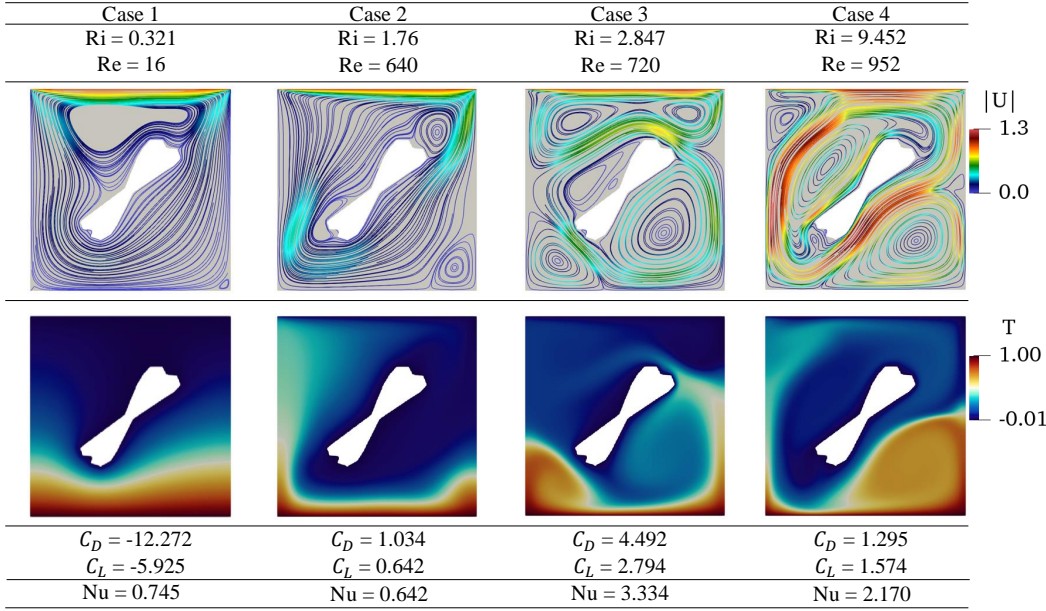

| | Case 1 | Case 2 | Case 3 | Case 4 |
|---|---|---|---|---|
| | Ri = 0.321 | Ri = 1.76 | Ri = 2.847 | Ri = 9.452 |
| | Re = 16 | Re = 640 | Re = 720 | Re = 952 |
| $C_D =$ | -12.272 | 1.034 | 4.492 | 1.295 |
| $C_L =$ | -5.925 | 0.642 | 2.794 | 1.574 |
| Nu = | 0.745 | 0.642 | 3.334 | 2.170 |

**Figure 10:** *Drag coefficients ($C_D$), lift coefficients ($C_L$), and Nusselt numbers (Nu) for different shapes and different Richardson numbers in NSHT LDC simulations (random Reynolds number).*

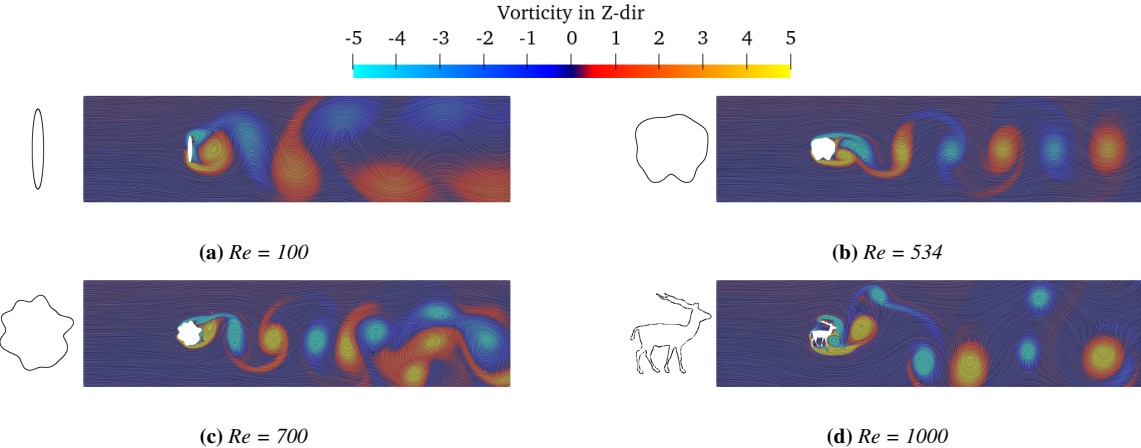

**(a)** *Re = 100*   **(b)** *Re = 534*

**(c)** *Re = 700*   **(d)** *Re = 1000*

**Figure 11:** *Time snapshots of vorticity for the FPO problem, illustrating vortex shedding profiles across four samples of geometries and Reynolds numbers. Each sample showcases distinct flow characteristics and vortex dynamics.*

2D simulation, we ensure that the mesh size is smaller than the Kolmogorov scale, calculated based on the Reynolds number, to capture the full range of turbulent motions.

The 3D LDC cases are similarly defined and simulated. In the 3D LDC setup, a cube features five stationary walls and one moving lid, with a domain size of [0, 2] × [0, 2] × [0, 2]. An object is placed in the middle of the flow within the chamber. Due to the computational expense of performing DNS on 3D geometries, we report 500 simulations of complex objects. Here, we focused on parametric shapes, specifically ellipsoids and tori, exhibiting various aspect ratios and orientations. Examples of LDC simulations showing streamlines and around the object is shown in Figure 12. As Reynolds number increase, more circulation regions around the object appear.

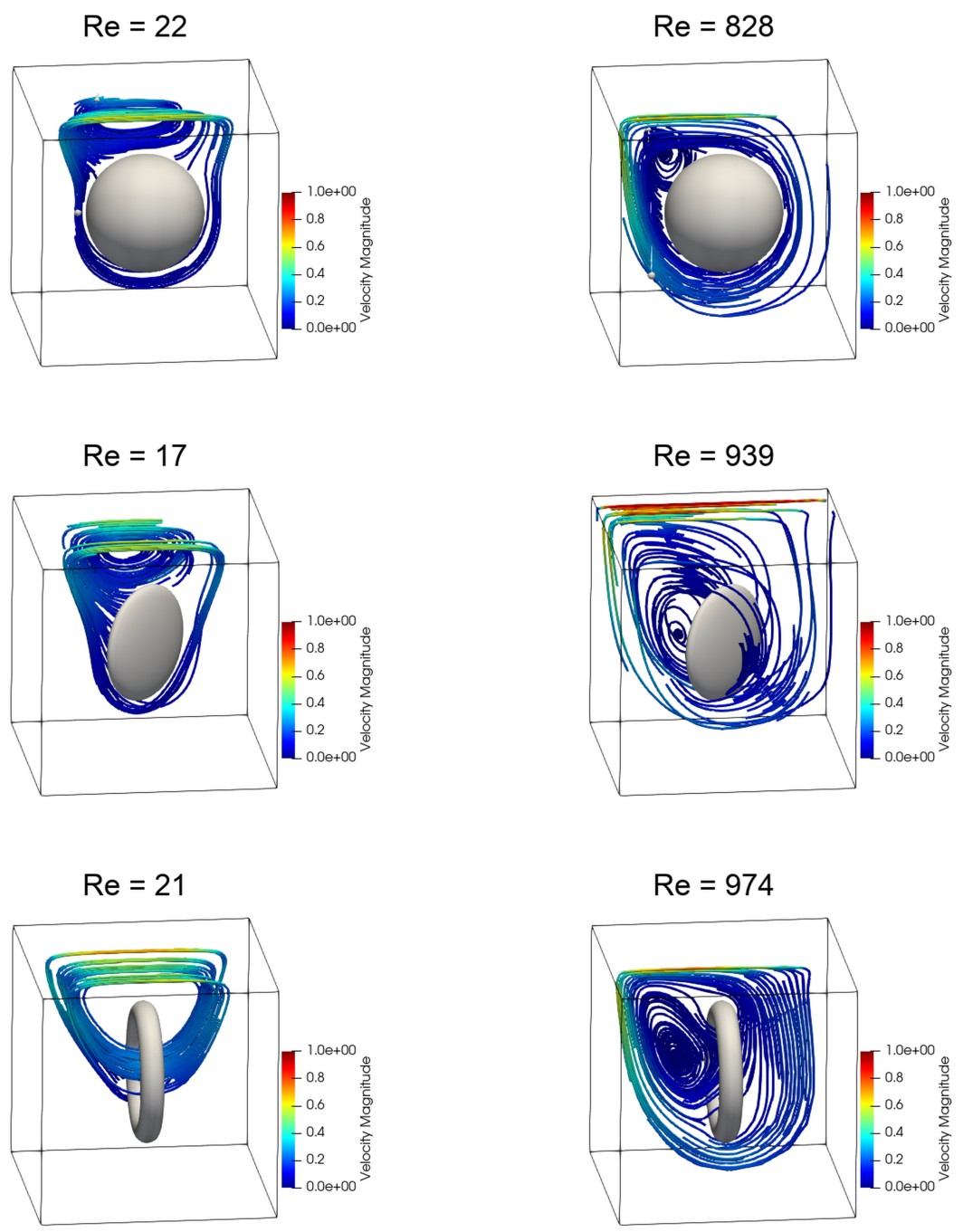

**Figure 12:** *Streamlines around a torus/ellipsoid for pure Navier-Stokes LDC simulations in 3D. Each row represents one object, with a low Reynolds number result on the left and a high Reynolds number result on the right.*

### 4.5 MetaData

**Input Fields:**  We have provisioned the following input fields:

1. **Reynolds Number** and **Grashof Number:** We feed the Reynolds and Grashof numbers as a concordant (matching the dimensions of the flow domain, e.g., for 2D lid-driven cavity problem, $512 \times 512$) array comprising of a single integer value everywhere.

2. **Geometry Mask:** A geometry mask ($g$) is a binary representation of a shape or object within a spatial domain. Each element in the mask can have one of two values, typically 0 or 1, where: 0 indicates that the point is outside the object and 1 indicates that the point is inside the object. The geometry mask helps identify and isolate the region of interest (the object) from the background or surrounding space. This information is again packaged as a concordant array with each entry marked as 0 or 1.

3. **Signed Distance Field (SDF):** A Signed Distance Field ($s$) is a scalar field that represents the shortest distance from any point in space to the surface of a given shape. The "signed" aspect of the signed distance field indicates whether the point is inside or outside the shape. A negative value indicates the point is inside the shape, while a positive value indicates the point is outside the shape. A value of zero indicates the point is exactly on the surface of the shape. The SDF provides additional information through the distance values, which is important for understanding the spatial relationship between locations and the geometric boundary. This method offers richer geometric information compared to the binary geometry mask, as it includes both positional and distance information. Yet again, this information is packaged as a concordant array, with each entry representing the nearest distance to the geometrical shape.

**Output Fields:**  We are interested in obtaining field solutions (i.e., solutions at every point in the interior of the domain) for certain cardinal fields. For a 2D solution domain, these are: $u$ - velocity in $x$ direction, $v$ - velocity in $y$ direction, $p$ - pressure and $\theta$ - temperature. Additionally, depending on whether we are solving a steady-state or a time-dependent problem, we would have either have one final snapshot of these cardinal fields or a sequence of these fields distributed uniformly over time.

**Resolution:**  We perform simulations at DNS resolution. FlowBench contains postprocessed simulation results at three different resolutions. This serves multiple purposes: First, it allows the community to systematically explore tradeoffs in the amount of data vs resolution vs accuracy. Second, since we perform non-interpolatory sub-sampling, this data allows the community to systematically test PDE super-resolution approaches. Finally, the lower-resolution datasets offer easier opportunities to train the data. For instance, at the time of submission, we could not train the 3D SciML models at the highest resolution available. For the steady state cases, we provide data at resolutions $512 \times 512$, $256 \times 256$, $128 \times 128$ ($\times 128$ - for 3D LDC NS). For the time dependant cases, we provide raw data at resolutions, $1024 \times 256$, and $512 \times 128$. We provide 240 snapshots for each time-dependant case. With only a light amount of postprocessing, the end user can choose from these 240 timesteps to create more managable datasets for machine learning, e.g., taking snapshots every sixth time step.

**Dataset Format:**  All of our datasets are provided as numpy compressed (`.npz`) files. For each steady-state problem, we provide two `.npz` files - one for input and one for output. The input files are suffixed with the marker `"_X.npz"`, and similarly, the corresponding output files are suffixed with the marker `"_Y.npz"`. In 2D, each of these `.npz` files contains a 4D `numpy` tensor of the following form:

$$[\textbf{samples}][\textbf{number\_of\_channels}][\textbf{resolution\_x}][\textbf{resolution\_y}]$$

Similarly, in 3D, each of these `.npz` files contains a 5D `numpy` tensor of the following form:

$$[\textbf{samples}][\textbf{number\_of\_channels}][\textbf{resolution\_x}][\textbf{resolution\_y}][\textbf{resolution\_z}]$$

For time-dependent problems, we provide a single file for each problem. This decision was taken to allow maximum flexibility to the end user in deciding what and how many time steps they want to use to train their models, as these time-dependent problems often take the shape of sequence-to-sequence formulations. In 2D, the resulting `.npz` files take the following form:

$$[\textbf{samples}][\textbf{number\_of\_time\_steps}][\textbf{number\_of\_channels}][\textbf{resolution\_x}][\textbf{resolution\_y}]$$

We provide a total of over 10K samples spread across four families of datasets. Table 9 in the Appendix provides a detailed formulaic description of the packaging of the input and output `numpy` tensors for each of these four families.

### 4.6 Evaluation Metrics and Test Datasets

**Errors:** We compute the following errors to evaluate the accuracy of all the models trained on FlowBench:
- *Mean Squared Error*:

$$\sum_{i=1}^{n}(y_i - y_i^{'})^2$$

- $L_2$ *Relative Error*:

$$L_2 = \sqrt{\frac{\sum_{i=1}^{n}(y_i - y_i^{'})^2}{\sum_{i=1}^{n} y_i^{'2}}}$$

- $L_\infty$ *Relative Error*:

$$L_\infty = \frac{\max \left|y_i - y_i^{'}\right|}{\max \left|y_i^{'}\right|}$$

*Note:* In all the above definitions, $y, y^{'}$ and $n$ indicate the predicted value from a model, the ground truth and the test dataset size respectively.

**Evaluation metrics:** We recommend a hierarchy of metrics ($M1, M2, M3, M4$) to comprehensively assess the performance of trained models using FlowBench:
- $M1$: *Global metrics*: Computing all three errors over the entire domain is a good primary metric to measure the accuracy of predicted velocity, pressure, and temperature fields, reflecting how closely the model's predictions match the true values.
- $M2$: *Boundary layer metrics*: The errors in velocity, pressure, and temperature fields over a narrow region around the object. We define the boundary layer by considering the solution conditioned on the Signed Distance Field ($0 \leq SDF \leq 0.2$). This tougher metric provides insight into the model's accuracy in predicting near-surface phenomena, which are crucial for applications including flow diagnostics, shape design, and dynamic control.
- $M3$: *Property metrics*: An application-driven metric is to evaluate the accuracy of the summary statistics – $C_D, C_L, Nu$. The coefficients of lift and drag are critical for assessing the forces acting on the object, while the average Nusselt number predicts heat transfer rates, all of which are key for flow-thermal management and engineering applications. This metric represents spatially averaged properties.
- $M4$: *Residual metrics*: One can also evaluate how well the predicted field satisfies the underlying (set of) PDE. Evaluating how well continuity ($\nabla \cdot u = 0$) is satisfied all over the domain is a measure of respecting the conservation of mass. Similarly, the global average of the PDE residual evaluates how well the model satisfies the underlying PDE in the domain. For a detailed explanation of the PDE residuals calculation, see Appendix B.3.

**Test Datasets:** We recommend evaluating trained models on a held out dataset using the standard 80-20 random split of the prepared dataset.

Together, these metrics and datasets provide a comprehensive evaluation framework, allowing practitioners to evaluate model accuracy, physical consistency, and practical reliability.

# 5 Experiments

We report baseline results for training a suite of the most common neural PDE solvers. We explored the following Neural Operators and Foundation Models, reporting results on the 2D LDC-NS: (a) Fourier Neural Operator (FNO) (Li et al., 2021), (b) DeepONet (Lu et al., 2021), (c) Poseidon (Herde et al., 2024). Plots of field solutions for $u$, $v$, and $p$, as well as training and validation losses, are available on our website. For training, we adhered closely to the published code examples. All the aforementioned models were trained on a single A100 80GB GPU using the Adam optimizer with a learning rate of $10^{-3}$ and were run for 400 epochs. The validation loss for all models converged and stabilized by 400 epochs.

**Table 5:** *The mean squared errors of various neural operators trained on the 2D LDC dataset. All errors are reported on the testing dataset.*

| Model | Geometry | M1 | M2 | M3 |
|---|---|---|---|---|
| **FNO** | G1 | $5.8 \times 10^{-3}$ | $7.1 \times 10^{-4}$ | $2.7 \times 10^{-3}$ |
| | G2 | $6.8 \times 10^{-3}$ | $4.9 \times 10^{-4}$ | $2.9 \times 10^{-3}$ |
| | G3 | $9.7 \times 10^{-3}$ | $1.2 \times 10^{-3}$ | $1.3 \times 10^{-2}$ |
| **DeepONet** | G1 | $4.9 \times 10^{-3}$ | $1.8 \times 10^{-3}$ | $1.8 \times 10^{-2}$ |
| | G2 | $2.3 \times 10^{-3}$ | $7.0 \times 10^{-4}$ | $7.0 \times 10^{-3}$ |
| | G3 | $6.0 \times 10^{-3}$ | $1.4 \times 10^{-3}$ | $3.7 \times 10^{-2}$ |
| **poseidon** | G1 | $2.2 \times 10^{-4}$ | $7.7 \times 10^{-5}$ | $5.2 \times 10^{-4}$ |
| | G2 | $\mathbf{1.4 \times 10^{-4}}$ | $\mathbf{4.9 \times 10^{-5}}$ | $\mathbf{4.1 \times 10^{-4}}$ |
| | G3 | $4.0 \times 10^{-4}$ | $1.4 \times 10^{-4}$ | $5.4 \times 10^{-3}$ |

**Table 6:** *The relative $L_2$ error of various neural operators trained on the 2D LDC dataset. All metrics are reported on the testing dataset.*

| Model | Geometry | M1 | M2 | M3 |
|---|---|---|---|---|
| **FNO** | G1 | $3.3 \times 10^{-1}$ | $3.6 \times 10^{-1}$ | $7.8 \times 10^{-1}$ |
| | G2 | $3.8 \times 10^{-1}$ | $4.3 \times 10^{-1}$ | $6.8 \times 10^{-1}$ |
| | G3 | $4.9 \times 10^{-1}$ | $6.0 \times 10^{-1}$ | $8.7 \times 10^{-1}$ |
| **DeepONet** | G1 | $3.9 \times 10^{-1}$ | $5.1 \times 10^{-1}$ | $1.4 \times 10^{0}$ |
| | G2 | $4.9 \times 10^{-1}$ | $5.7 \times 10^{-1}$ | $8.4 \times 10^{-1}$ |
| | G3 | $6.2 \times 10^{-1}$ | $7.1 \times 10^{-1}$ | $1.4 \times 10^{0}$ |
| **poseidon** | G1 | $\mathbf{1.9 \times 10^{-1}}$ | $\mathbf{2.7 \times 10^{-1}}$ | $\mathbf{2.5 \times 10^{-1}}$ |
| | G2 | $2.4 \times 10^{-1}$ | $\mathbf{2.7 \times 10^{-1}}$ | $3.7 \times 10^{-1}$ |
| | G3 | $3.9 \times 10^{-1}$ | $4.9 \times 10^{-1}$ | $3.7 \times 10^{-1}$ |

**Table 7:** *The relative $L_\infty$ error of various neural operators trained on the 2D LDC dataset. All metrics are reported on the testing dataset.*

| Model | Geometry | M1 | M2 | M3 |
|---|---|---|---|---|
| **FNO** | G1 | $6.1 \times 10^{-1}$ | $7.2 \times 10^{-1}$ | $9.8 \times 10^{-1}$ |
| | G2 | $7.4 \times 10^{-1}$ | $8.1 \times 10^{-1}$ | $8.5 \times 10^{-1}$ |
| | G3 | $9.8 \times 10^{-1}$ | $1.1 \times 10^{0}$ | $1.2 \times 10^{0}$ |
| **DeepONet** | G1 | $8.3 \times 10^{-1}$ | $9.1 \times 10^{-1}$ | $3.3 \times 10^{0}$ |
| | G2 | $9.0 \times 10^{-1}$ | $1.0 \times 10^{0}$ | $2.6 \times 10^{0}$ |
| | G3 | $1.1 \times 10^{0}$ | $1.3 \times 10^{0}$ | $3.1 \times 10^{0}$ |
| **poseidon** | G1 | $\mathbf{3.3 \times 10^{-1}}$ | $\mathbf{4.3 \times 10^{-1}}$ | $\mathbf{2.9 \times 10^{-1}}$ |
| | G2 | $4.1 \times 10^{-1}$ | $5.7 \times 10^{-1}$ | $4.5 \times 10^{-1}$ |
| | G3 | $6.4 \times 10^{-1}$ | $7.7 \times 10^{-1}$ | $5.1 \times 10^{-1}$ |

Table 5 through Table 7 present a comprehensive evaluation of the performance of all the trained models. The models were assessed using the four metrics outlined in Section 4.6, across each geometry class and at the highest resolution ($512 \times 512$) for all seven models. Our findings indicate that in terms of both *mean squared error* and $L_\infty$ error, the foundation models consistently outperformed the neural operators across all geometries and metrics. For the $L_2$ error, the neural operators performed comparably to the foundation models. We also include examples of the field prediction for $u$, $v$ and $p$ using the three scientific machine learning models in Appendix B.4. We encourage further exploration and validation of these findings by the research community.

**Table 8:** *The residual errors (M4) of various neural operators trained on the 2D LDC dataset. All metrics are reported on the testing dataset. Errors reported on the test dataset (per sample) normalized by the total number of mesh points.*

| Model | Geometry | momentum | continuity |
|---|---|---|---|
| **FNO** | G1 | $1.03 \times 10^{-4}$ | $9.11 \times 10^{-8}$ |
| | G2 | $5.80 \times 10^{-5}$ | $7.05 \times 10^{-8}$ |
| | G3 | $7.62 \times 10^{-5}$ | $\mathbf{6.76 \times 10^{-8}}$ |
| **DeepONet** | G1 | $5.09 \times 10^{-5}$ | $8.96 \times 10^{-8}$ |
| | G2 | $\mathbf{4.51 \times 10^{-5}}$ | $1.39 \times 10^{-7}$ |
| | G3 | $5.72 \times 10^{-5}$ | $5.07 \times 10^{-7}$ |
| **poseidon** | G1 | $9.62 \times 10^{-5}$ | $1.35 \times 10^{-7}$ |
| | G2 | $8.35 \times 10^{-5}$ | $1.39 \times 10^{-7}$ |
| | G3 | $9.64 \times 10^{-5}$ | $1.46 \times 10^{-7}$ |

## 6 Conclusions

We introduce a comprehensive benchmark dataset designed for evaluating neural solvers of flow simulations over complex geometries. FlowBench encompasses 2D and 3D simulations, covering many scenarios, from steady-state problems to time-dependent problems. FlowBench aims to help develop scientific machine learning models by offering a complex dataset designed to challenge and benchmark their performance. Neural PDE solvers that can account for the effect of complex geometries can have a major impact on various applications ranging from bioengineering and power production to automotive and aerospace engineering defined by the interaction of complex geometrical objects with a fluid medium.

As a step toward reproducibility and ease of use, we have released an end-to-end tutorial in our follow-up work (Rabeh et al., 2024a) and an accompanying GitHub repository available (here). This repository provides step-by-step instructions and code for training and evaluating Fourier Neural Operator, DeepONet, and the pretrained poseidon model on the 2D LDC subset of FlowBench.

**Limitations**: (1) Our evaluation of existing neural PDE solvers is limited to one of the four FlowBench datasets and on seven neural PDE models. We encourage the community to contribute by evaluating a wider range of approaches using this dataset and the proposed metrics. Additionally, we plan to expand FlowBench with more data—particularly for 3D simulations—and invite the CFD community to do the same, extending the dataset to cover higher $Re$ and $Gr$ operating conditions.

## Acknowledgements

We gratefully acknowledge support from the NAIRR pilot program for computational access to TACC Frontera. This work is supported by the AI Research Institutes program supported by NSF and USDA-NIFA under AI Institute: for Resilient Agriculture, Award No. 2021-67021-35329. We also acknowledge partial support through NSF 2053760.

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

## Appendix A. Details of the CFD simulation framework

Our CFD framework is a well-validated and massively scalable software suite that uses a variant of the immersed boundary method integrated with octree meshes to perform highly efficient and accurate Large-Eddy Simulations (LES) of flows around complex geometries. This framework demonstrates scalability of up to 32,000 processors, achieved through several key innovations, including (a) Rapid in-out tests: These tests quickly determine whether a point is inside or outside a given geometry, significantly speeding up the simulation process. (b) adaptive quadrature: This technique ensures accurate force evaluation by dynamically adjusting the numerical integration based on the local complexity of the geometry, and (c) Tensorized operations: These operations optimize performance by leveraging tensor algebra for computational efficiency. Additional details of the CFD framework, including implementation, are provided in Saurabh et al. (2021a, 2023).

Our datasets and code are licensed under CC-BY-NC-4.0. Next, we briefly detail the mathematics of the shifted boundary method.

### A.1 Heuristic derivation of the SBM formulation for Navier-Stokes

A.1.1 DERIVATION FOR NITSCHE'S METHOD FOR NAVIER-STOKES

Nitsche's method starts with the Lagrange multiplier method and adds a penalty term to obtain the augmented Lagrangian formulation. We begin with the Lagrange multiplier method, which is given by:

$$\underbrace{\int w\lambda \, d\Gamma}_{consistency\ term} + \underbrace{\int \delta\lambda(u - u_D) \, d\Gamma}_{adjoint\text{-}consistency\ term}. \tag{7}$$

After adding the penalty term $\frac{\beta}{h \cdot Re} \int w(u - u_D) \, d\Gamma$, we can obtain Nitsche's method:

$$\underbrace{\int w\lambda \, d\Gamma}_{consistency\ term} + \underbrace{\int \delta\lambda(u - u_D) \, d\Gamma}_{adjoint\text{-}consistency\ term} + \underbrace{\frac{\beta}{h \cdot Re} \int w(u - u_D) \, d\Gamma}_{penalty\ term}. \tag{8}$$

Our objective is to determine the Lagrange multiplier, $\lambda$, for the Navier-Stokes equation. This is achieved by ensuring that the consistency term matches the boundary term, which arises when integration by parts is applied to the weak form of the Navier-Stokes equation. To derive the weak form, we start from the strong form of the Navier-Stokes equation and utilize test functions $w$ and $q$:

$$\begin{cases} \frac{\partial u}{\partial t} + u \cdot \nabla u - f - \nabla \cdot \sigma = 0 \\ \nabla \cdot u = 0 \end{cases}$$
$$\Rightarrow \int w\left(\frac{\partial u}{\partial t} + u \cdot \nabla u - f\right) d\Omega \underbrace{- \int w\nabla \cdot \sigma \, d\Omega}_{Cauchy\ stress\ term} + \int q\nabla \cdot u \, d\Omega = 0. \tag{9}$$

Performing integration by parts (Section A.1.2) on the Cauchy stress term, we obtain the following:

$$-\int w\nabla \cdot \sigma \, d\Omega = \underbrace{\int \nabla w : \sigma \, d\Omega}_{volume\ term} - \underbrace{\int w\sigma \cdot \mathbf{n} \, d\Gamma}_{boundary\ term}. \tag{10}$$

The boundary term is the same as the consistency term in Nitsche's formulation so that we can obtain the following:

$$\int w\lambda \, d\Gamma = -\int w\sigma \cdot \mathbf{n} \, d\Gamma \Rightarrow \lambda = -\sigma \cdot \mathbf{n} = -(\frac{2}{Re}\nabla^s u - pI) \cdot \mathbf{n}. \tag{11}$$

And, we can obtain $\delta\lambda = -(\frac{2}{Re}\nabla^s w + qI) \cdot \mathbf{n}$.
Upon substitution of $\lambda$ and $\delta\lambda$ into equation Equation 8, Nitsche's formulation for Navier-Stokes can be obtained, and is expressed as follows:

$$\underbrace{-\int w(\frac{2}{Re}\nabla^s u - pI) \, d\Gamma}_{\text{consistency term}} \underbrace{-\int (\frac{2}{Re}\nabla^s w + qI)(u - u_D) \, d\Gamma}_{\text{adjoint-consistency term}} + \underbrace{\frac{\beta}{h \cdot Re}\int w(u - u_D) \, d\Gamma}_{\text{penalty term}}. \tag{12}$$

### A.1.2 INTEGRATION BY PART FOR CAUCHY STRESS TERM

Doing integration by part for the Cauchy stress term in Equation 9, we write the term in the form of Einstein notation so that it is easier to calculate:

$$-\int w\nabla \cdot \sigma \, d\Omega = -\int w_i \frac{\partial \sigma_{ik}}{\partial x_k} \, d\Omega = -\int w_i \left( \frac{1}{Re}\frac{\partial \varepsilon_{ik}}{\partial x_k} - \delta_{ik}\frac{\partial p}{\partial x_k} \right) d\Omega = -\int w_i \left( \frac{1}{Re}\frac{\partial \varepsilon_{ik}}{\partial x_k} - \frac{\partial p}{\partial x_i} \right) d\Omega$$

$$= \int \left( \underbrace{-\frac{1}{Re}\frac{\partial(w_i\varepsilon_{ik})}{\partial x_k}}_{\text{term 1}} + \underbrace{\frac{1}{Re}\varepsilon_{ik}\frac{\partial w_i}{\partial x_k}}_{\text{term 2}} + \underbrace{\frac{\partial(pw_i)}{\partial x_i}}_{\text{term 1}} - \underbrace{p\frac{\partial w_i}{\partial x_i}}_{\text{term 2}} \right) d\Omega \tag{13}$$

$$= \underbrace{\int \left( -\frac{1}{Re}\frac{\partial(w_i\varepsilon_{ik})}{\partial x_k} + \frac{\partial(pw_i)}{\partial x_i} \right) d\Omega}_{\text{term 1}} + \underbrace{\int \left( \frac{1}{Re}\varepsilon_{ik}\frac{\partial w_i}{\partial x_k} - p\frac{\partial w_i}{\partial x_i} \right) d\Omega}_{\text{term 2}}$$

$$= \int w_i(-\frac{1}{Re}\varepsilon_{ik}n_k + pn_i) \, d\Gamma + \int \frac{\partial w_i}{\partial x_k}(\frac{1}{Re}\varepsilon_{ik} - p\delta_{ik}) \, d\Omega = \int w_i(-\sigma_{ik}n_k) \, d\Gamma + \int \frac{\partial w_i}{\partial x_k}\sigma_{ik} \, d\Omega$$

$$= -\int w\sigma \cdot \mathbf{n} \, d\Gamma + \int \nabla w : \sigma \, d\Omega. \tag{14}$$

### A.1.3 SBM FOR NAVIER-STOKES

In the SBM, instead of enforcing boundary condition on the true boundary ($\Gamma$), we enforce Nitsche's method on the surrogate boundary ($\tilde{\Gamma}$) by changing the boundary condition from $u_D$ to $\tilde{u}_D$:

$$\underbrace{-\int w(\frac{2}{Re}\nabla^s u - pI) \, d\tilde{\Gamma}}_{\text{consistency term}} \underbrace{-\int (\frac{2}{Re}\nabla^s w + qI)(u - \tilde{u}_D) \, d\tilde{\Gamma}}_{\text{adjoint-consistency term}} + \underbrace{\frac{\beta}{h \cdot Re}\int \tilde{w}(u - \tilde{u}_D) \, d\tilde{\Gamma}}_{\text{penalty term}}. \tag{15}$$

With Taylor expansion to account for the discrepancy between the surrogate and true boundary, the boundary condition becomes $\tilde{u}_D = u_D - \nabla u \cdot d$. Substituting this expression into Equation 15, we obtain:

$$\underbrace{-\int w(\frac{2}{Re}\nabla^s u - pI) \, d\tilde{\Gamma}}_{\text{consistency term}} \underbrace{-\int (\frac{2}{Re}\nabla^s w + qI)(u + \nabla u \cdot d - u_D) \, d\tilde{\Gamma}}_{\text{adjoint-consistency term}}$$

$$+ \underbrace{\frac{\beta}{h \cdot Re}\int (w + \nabla w \cdot d)(u + \nabla u \cdot d - u_D) \, d\tilde{\Gamma}}_{\text{penalty term}}. \tag{16}$$

### A.2 Solving the CFD equations: Automated creation of meshes involving complex geometries

Tree-based mesh generation, using quadtrees in 2D and octrees in 3D, is common in computational sciences due to its simplicity and parallel scalability. These tree-based data structures enable efficient refinement of regions of interest, facilitating their deployment in large-scale multi-physics simulations. Our mesh generation tool, Dendro-kt (Saurabh et al., 2021b; Heisler et al., 2023; Gamdha et al., 2023), provides balanced, partitioned, and parallel tree structures, making it highly effective for large-scale numerical PDE discretizations.

**Lid-driven cavity (LDC) flow:** For the LDC case, we use an octree-based mesh generation framework to create a uniform mesh with an element size of $\frac{1}{2^9}$ over a $[0, 2] \times [0, 2]$ domain. This produces a $512 \times 512$ mesh resulting in total degress-of-freedom of $512 \times 512 \times 3 \sim 750K$.

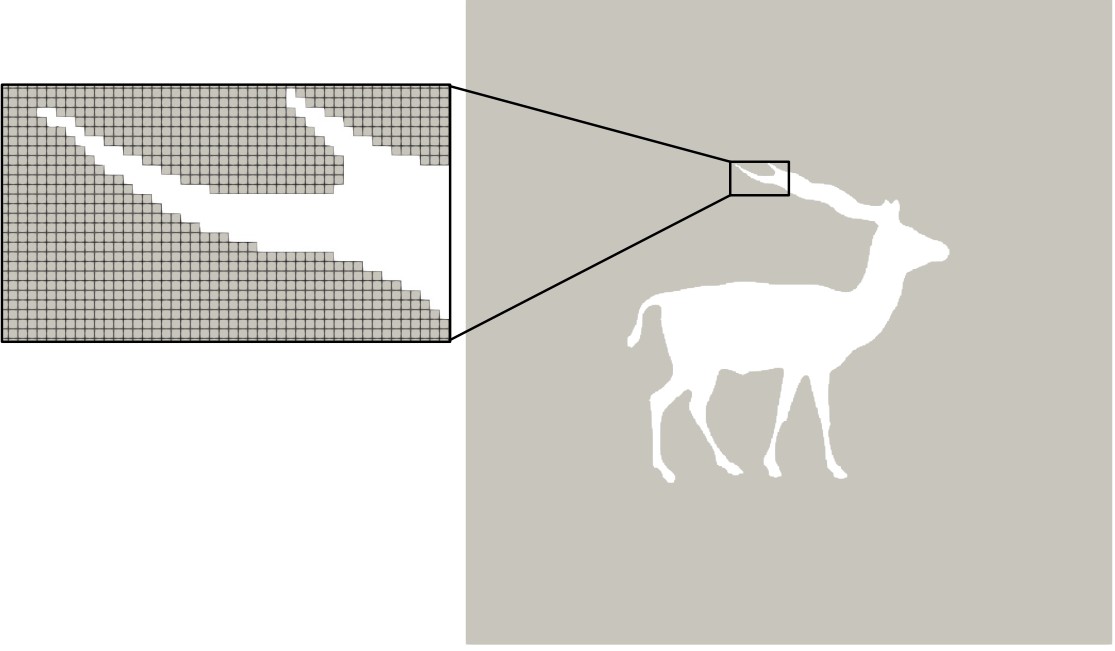

**Figure 13:** *Uniform octree mesh for the lid-driven cavity flow. We illustrate the mesh with a representative shape from the Skelneton data. This mesh is carved out from a $512 \times 512$ uniform tessellation of the domain.*

**Flow past object (FPO):** Our computational domain is a rectangular region spanning $[0, 64] \times [0, 16]$, with the complex geometry centered at the coordinates $(6, 8)$. Our mesh refinement strategy involves several layers of progressively coarser mesh surrounding the geometry and along the wake. Specifically, we utilize five concentric circles centered at $(6, 8)$, with the innermost circle having a radius of 0.71 and a refinement level of 13 (yielding an element size of $\frac{64}{2^{13}}$), the next circle with a radius of 0.8 and a refinement level of 12 ($\frac{64}{2^{12}}$ element size), the third circle with a radius of 1 and a refinement level of 11 ($\frac{64}{2^{11}}$ element size), the fourth circle with a radius of 2.5 and a refinement level of 10 ($\frac{64}{2^{10}}$ element size), and the outermost circle with a radius of 3 and a refinement level of 9 ($\frac{64}{2^9}$ element size). We use a non-dimensional time step of 0.01 for the simulation. Starting from a non-dimensional total time of 392, we begin outputting results every 0.05 non-dimensional time units until reaching a non-dimensional total time of 404. The period from non-dimensional time 392 to 404 is when we output results that are post-processed for use in FlowBench. This time step was selected to ensure that each vortex shedding cycle is captured with at least 100 snapshots, based on an evaluation of the shedding frequencies across the range of Reynolds numbers and shapes. Consequently, each sample contains at least two complete shedding cycles.

Additionally, we define two rectangular refinement regions aligned with the flow direction. The first rectangle has its bottom-left corner at $(6, 5.5)$ and top-right corner at $(64, 10.5)$, with a refinement level of 10 ($\frac{64}{2^{10}}$

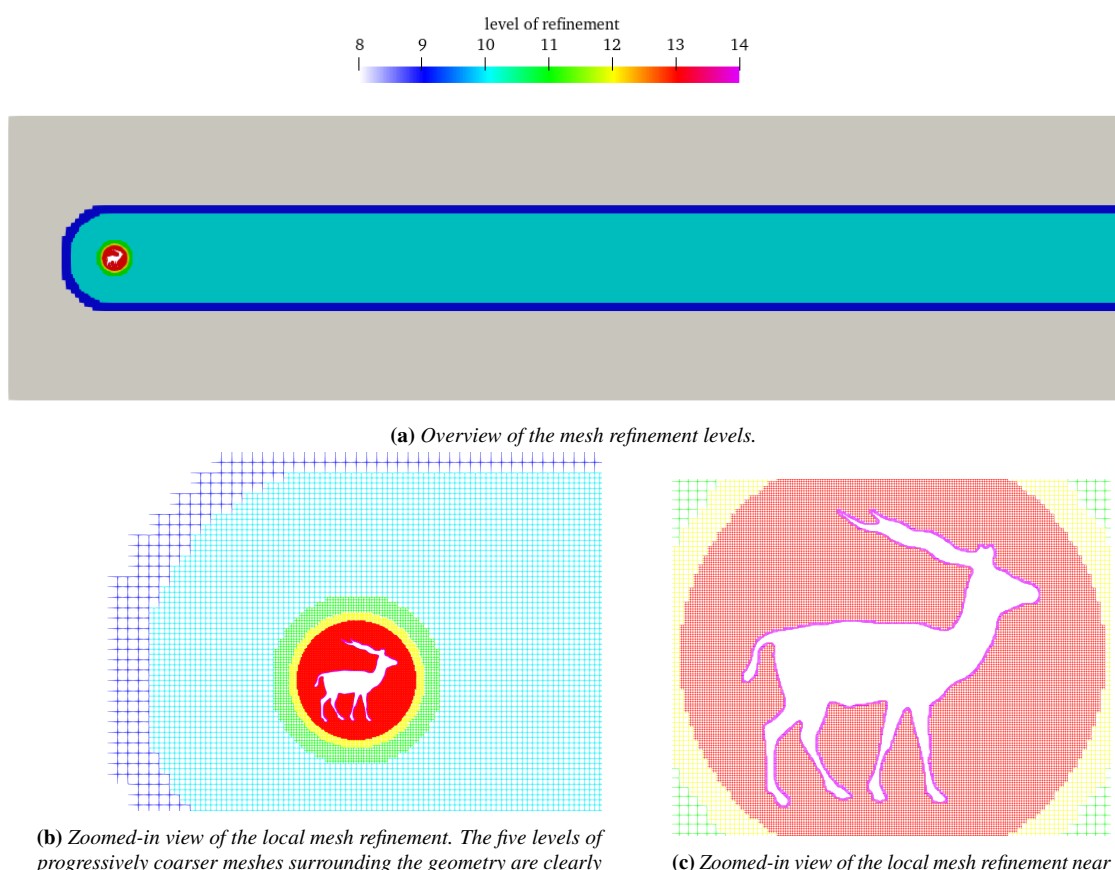

**(a)** *Overview of the mesh refinement levels.*

**(b)** *Zoomed-in view of the local mesh refinement. The five levels of progressively coarser meshes surrounding the geometry are clearly visible.*

**(c)** *Zoomed-in view of the local mesh refinement near the geometry.*

**Figure 14:** *Local refinement octree mesh for the flow past the geometry. The geometry is represented by a deer.*

element size). The second rectangle has its bottom-left corner at $(6, 5)$ and top-right corner at $(64, 11)$, with a refinement level of 9 ($\frac{64}{2^9}$ element size). Close to the geometry, to ensure detailed capture of the flow dynamics, we achieve a finer mesh with a refinement level of 14, resulting in an element size of $\frac{64}{2^{14}}$. The mesh with different refinement levels are shown in Figure 14.

The refinement strategy leads to a total nodal point number of 117978. Consequently, the total degrees of freedom (DOFs) amount is $117978 \times 3$ (353934). The combination of adaptive refinement and SBM formulation is critical to our ability to solve the PDEs in complex geometries. For comparison, if we were to use a uniform mesh (as in the case of the LDC) instead of an adaptively refined mesh, we would have had a problem with $2^{14} \times 2^{12} \times 3 \sim 201M$ DOFs!

**3D LDC:** We use a similar adaptive refinement strategy for the 3D LDC case. See Figure 15 for an example of an object and the mesh used. Close to the object's surface, we use a refinement of level 9, producing elements of size $2/2^9$, and away from the object, we progressively coarsen the mesh to a refinement level of 7, for elements of size $2/2^7$. This produces a problem with $2.5M$ DOFs. For the boundary condition, a uniform velocity is applied along one direction to simulate the lid's movement, while the remaining two velocity components are set to zero. Specifically, the top boundary of the cavity moves with a velocity of $u = 1$, while the transverse components $v = 0$ and $w = 0$, indicating no movement in those directions. All other boundaries are treated as stationary walls, where the no-slip condition applies, meaning $u = v = w = 0$. An illustration of the boundary conditions for the 3D lid-driven cavity (LDC) is shown in Figure 16.

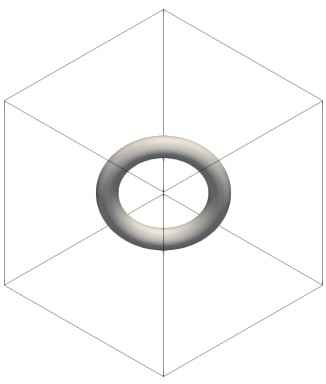
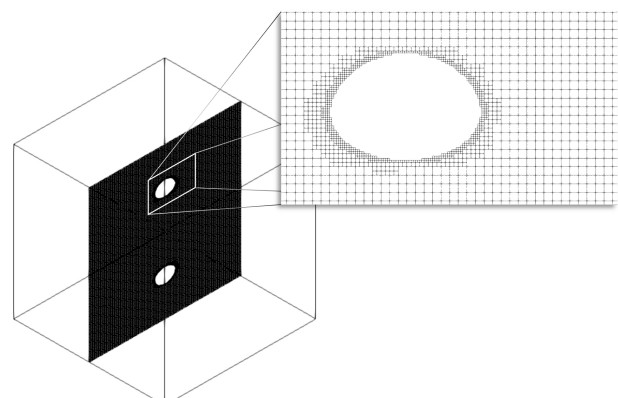

**(a)** *The object inside the lid driven cavity domain.*    **(b)** *Octree mesh showing local refinement near the object boundary.*

**Figure 15:** *An example shape in the 3D LDC case, along with a slice of the computational mesh*

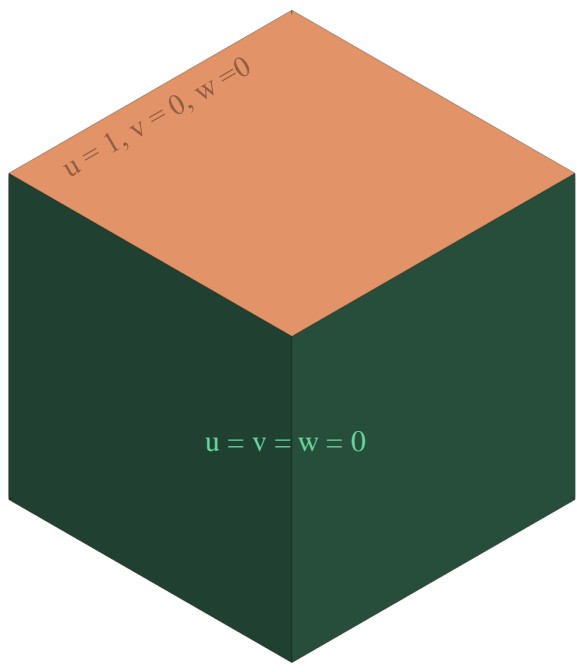

**Figure 16:** *Boundary conditions for the 3D lid-driven cavity flow. The top boundary has $u = 1, v = 0, w = 0$, while all other boundaries are no-slip.*

**Solver configuration:**    We use the Petsc linear algebra package for solving the system of equations. We utilize the BCGS solver, with an ASM pre-conditioner. The solver continues iterating until a relative tolerance, $rtol$, of $10^{-8}$, is reached.

### A.3  Postprocessing of the results to compute force coefficients and Nusselt number

After the CFD solve, we also compute three engineering summary variables. The drag coefficient ($C_D$) represents the non-dimensional force exerted on an object in the direction of the flow. It is calculated using the

formula:

$$C_D = \frac{2F_x}{A_{py}}.$$ (17)

where $F_x$ is the drag force and $A_{py}$ is the projection area in the y-direction.

The lift coefficient ($C_L$) indicates the non-dimensional lift force acting perpendicular to the flow direction. It is defined as:

$$C_L = \frac{2F_y}{A_{px}}.$$ (18)

where $F_y$ is the lift force and $A_{px}$ is the projection area in the x-direction.

Finally, we compute the Nusselt number to determine the amount of heat transfer. The local Nusselt number can be defined as:

$$Nu = \nabla T \cdot \mathbf{n}.$$ (19)

The averaged Nusselt number can be written as:

$$\overline{Nu} = \frac{\int Nu \, d\Gamma}{\int d\Gamma}.$$ (20)

We compute the average $Nu$ across both the bottom wall as well as the surface of the object.

## Appendix B. Benchmark Setup

### B.1 Downsampling

We used our in-house FEM simulation software `DendroKT` to perform forward simulations. In order to capture the physics precisely, we use a very fine resolution – with mesh sizes at the Kolmogorov length scale in 2D, and mesh sizes at $2\times$ the Kolmogorov length scale in 3D – to run the FEM simulations. While this approach allows us to model the physics very precisely, the resulting tensors generated at those resolutions are impractically large in size i.e., running into hundreds of gigabytes. We therefore use the ParaView tool (Ayachit, 2015) to downsample the original resolution to lower resolutions such as, $512 \times 512$, $256 \times 256$, $128 \times 128 \times 128$ and $1024 \times 256$. It is important to note that this downsampling is on the fully resolved data, and thus still captures all the larger scale features (as well as the impact of the small scale features on the large scale features). Note that for the 3D LDC cases, although the forward simulations are performed on an adaptive mesh with refinements up to level-9 near the object and level-7 elsewhere, the final processed results (the .npz tensors) are uniformly downsampled to a uniform $128 \times 128 \times 128$ resolution.

### B.2 Machine Learning

Our Machine Learning pipeline consists of three modules. In the first module, we undertake the tensorization exercise. We group the available downsampled data to a four or five dimensional *numpy* tensor depending on the nature of the physics outlined in Table 9. Tensorization helps us to subset the data using the tensor notation, and it also makes the data ready to be fed into Neural Operators/Foundation models. Data at this point is fed into the second module, where we primarily train the Neural Operators/Foundation Models. We have two principal objectives in this module, a) Learn the best hyperparameters for a given dataset and a given Neural Operator/Foundation Model. b) Select the best hyperparameter and perform a final training exercise to record the performance of each Neural Operator/Foundation Model on every dataset. Finally, we move to the post-processing module, where we report a complete suite of goodness of fit statistics using the trained model on a held-out data sample. We make our code, for select Neural Operators, publicly available.

**Table 9:** *Formulaic description of the input and output tensors. 3000/6000/1150/500 are sample sizes for the dataset. 240 is the number of equi-spaced time snapshots for the FPO case; $x, y(, z)$ are the dimensions of a field. E.g., For LDC - NS $Y[0, 1, :, :]$ indicates the pointwise $v$ velocity over the entire grid for the first sample in the dataset. As stated in*  *the FPO data requires some light postprocessing to bring it to the desired* `numpy` *tensor format.*

| Dataset | Dim. | Input Tensor | Output Tensor |
|---|---|---|---|
| LDC - NS | 2 | $X[3000][Re, g, s][x][y]$ | $Y[3000][u, v, p][x][y]$ |
| LDC - NS+HT | 2 | $X[5990][Re, Gr, g, s][x][y]$ | $Y[5990][u, v, p, \theta][x][y]$ |
| FPO - NS | 2 | $X[1150][Re, g, s][x][y]$ | $Y[1150][240][u, v, p][x][y]$ |
| LDC - NS | 3 | $X[500][Re, g, s][x][y][z]$ | $Y[500][u, v, p][x][y][z]$ |

## B.3 Residual Metrics (M4)

One can also evaluate how well the predicted field satisfies the underlying (set of) PDE. Evaluating how well continuity ($\nabla \cdot \mathbf{u} = 0$) is satisfied over the domain provides a measure of mass conservation. Similarly, the global average of the PDE residual evaluates how well the model satisfies the underlying PDE in the domain. To compute these metrics, we adopt a finite element-based approach. For each element $e$ in the finite element mesh $\mathcal{K}_h$, the local residuals for the Navier-Stokes momentum equations in the $x$- and $y$-directions are computed as:

$$r_{\mathrm{x}} = \frac{\partial u_x}{\partial t} + u \cdot \nabla u_x - \eta \nabla^2 u_x + \frac{\partial p}{\partial x},$$

$$r_{\mathrm{y}} = \frac{\partial u_y}{\partial t} + u \cdot \nabla u_y - \eta \nabla^2 u_y + \frac{\partial p}{\partial y}.$$

These residuals are evaluated at several quadrature points (using a 2×2 Gauss quadrature in 2D) via linear basis function interpolation. The $L_2$ norms of the local residuals for each element are computed as:

$$\|r_{\mathrm{x}}\|_{L_2(e)} = \left( \int_e r_{\mathrm{x}}^2 \, de \right)^{1/2}, \quad \|r_{\mathrm{y}}\|_{L_2(e)} = \left( \int_e r_{\mathrm{y}}^2 \, de \right)^{1/2}.$$

The total PDE residual is obtained by summing the squared contributions over all elements:

$$M4 \text{ (momentum)} = r_{\text{Total}} = \sum_{e \in \mathcal{K}_h} \left( \|r_{\mathrm{x}}\|_{L_2(e)}^2 + \|r_{\mathrm{y}}\|_{L_2(e)}^2 \right).$$

In addition, the continuity residual is computed to enforce the incompressibility condition. Specifically, the residual of the continuity equation,

$$\nabla \cdot \mathbf{u} = u_x + v_y + w_Z,$$

is evaluated at the same quadrature points. The element-wise continuity residual is calculated as:

$$\|r_c\|_{L_2(e)} = \left( \int_e (\nabla \cdot \mathbf{u})^2 \, de \right)^{1/2},$$

and the global continuity residual is given by:

$$M4 \text{ (continuity)} = r_{c,\text{Total}} = \sum_{e \in \mathcal{K}_h} \|r_c\|_{L_2(e)}^2.$$

## B.4 Field Predictions

Below, we present the field predictions for a representative sample using scientific machine learning models (FNO, DeepONet, Poseidon-T). The top row illustrates the ground truth fields for $u$, $v$, and $p$. The middle row displays the model predictions, while the bottom row depicts the corresponding error fields, defined as the difference between the predictions and the ground truth.

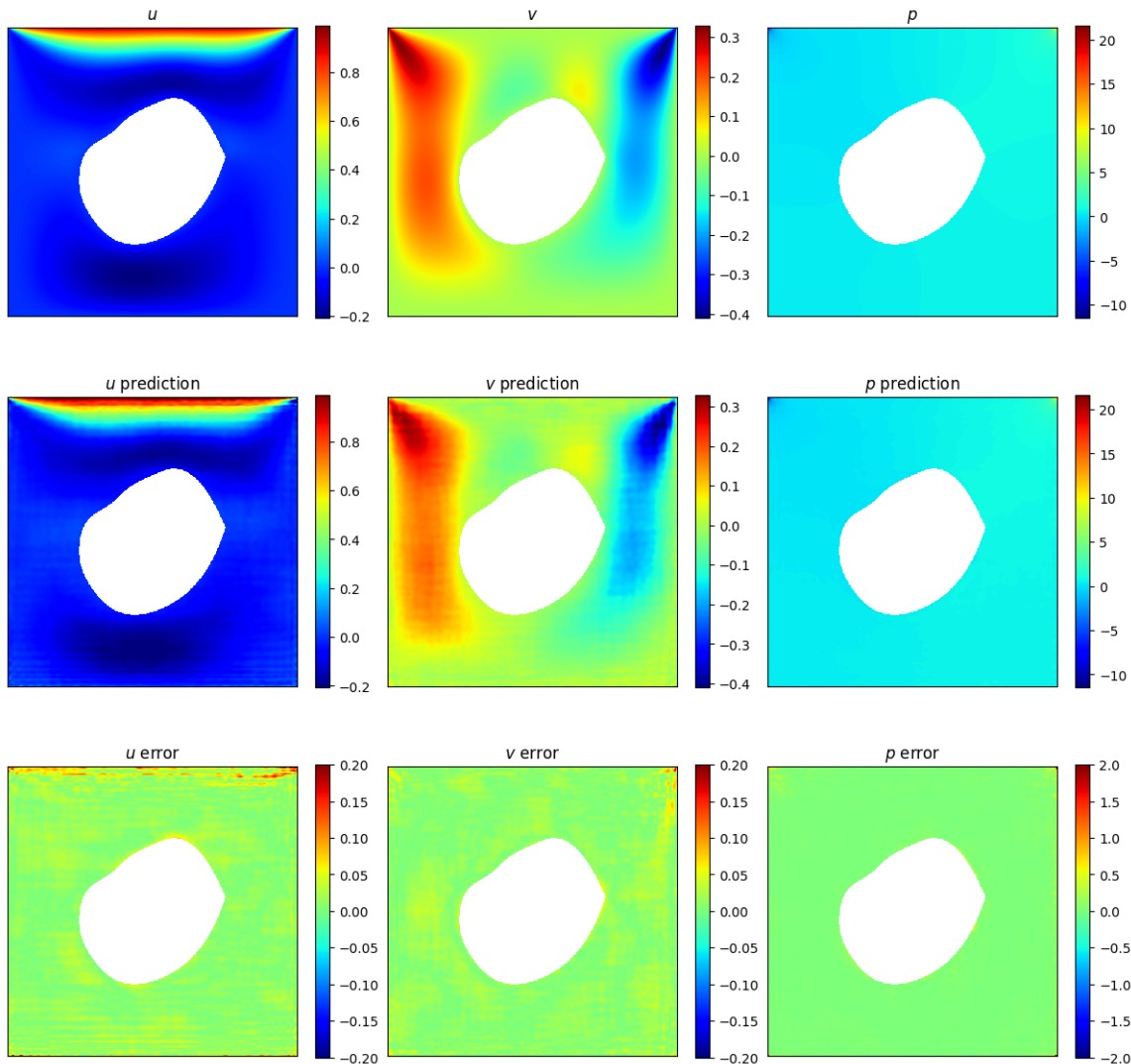

**Figure 17:** *Fourier Neural Operator: Prediction vs. Ground Truth and Error for $u, v, p$.*

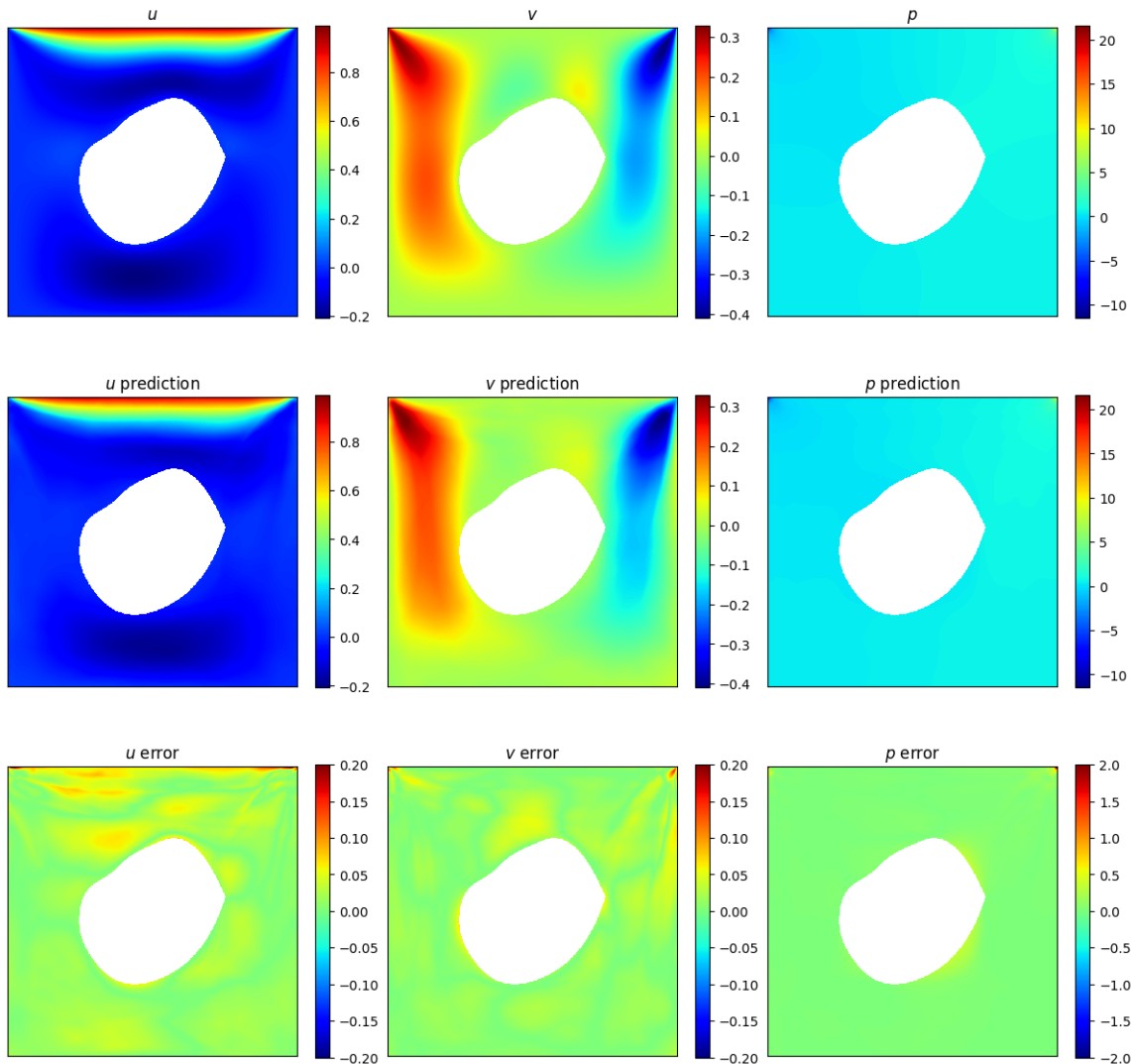

**Figure 18:** *DeepONet: Prediction vs. Ground Truth and Error for $u, v, p$.*

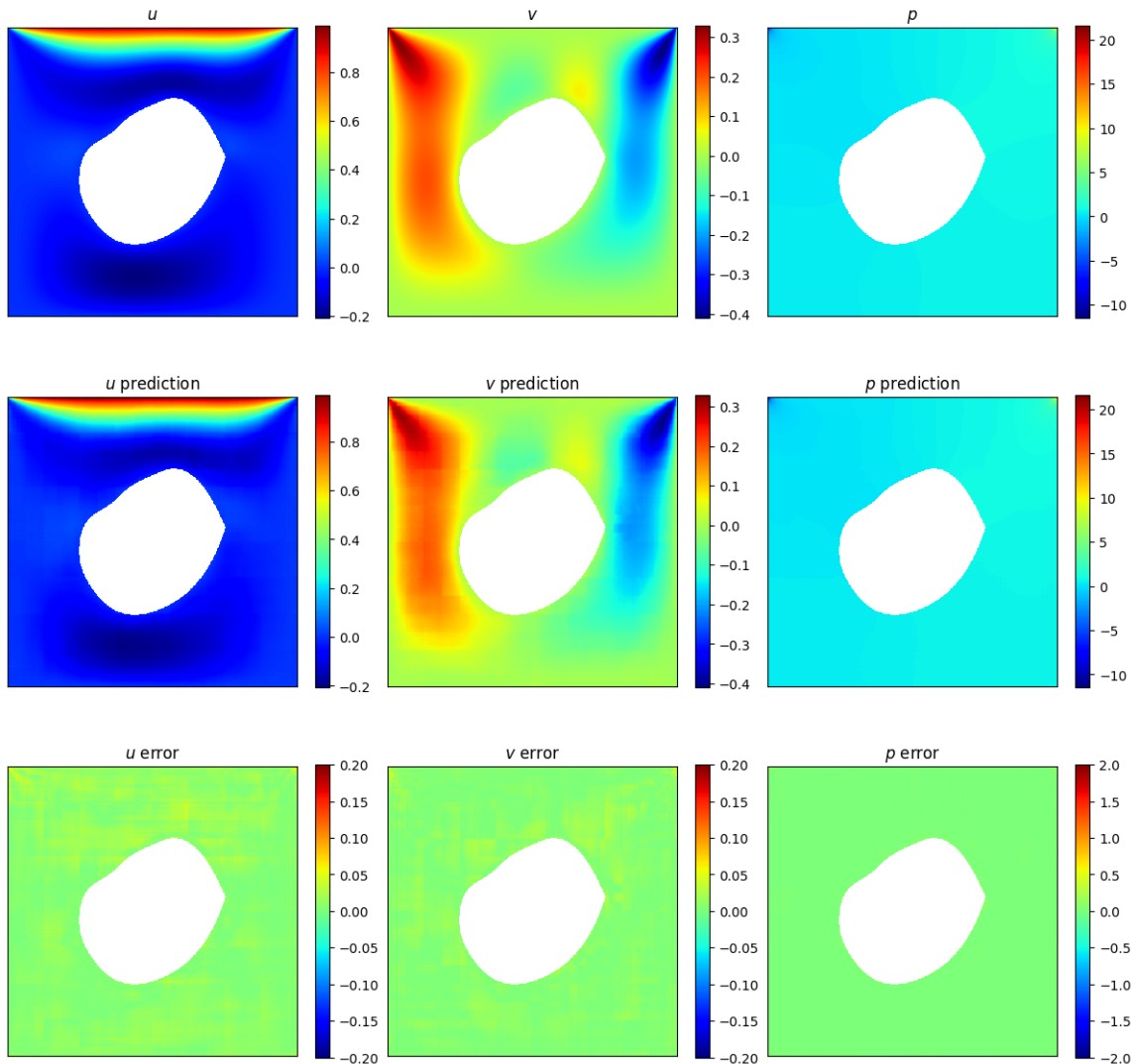

**Figure 19:** *poseidon-T: Prediction vs. Ground Truth and Error for $u, v, p$.*

