# OpenReview forum: "FlowBench: A Large Scale Benchmark for Flow Simulation over Complex Geometries"
_DMLR — Accepted by DMLR_

### Review · Reviewer_H2LD · 2025-02-02

**Recommendation:** 3
**Confidence:** 2

**Summary Of Contributions:**

The paper introduces FlowBench, a large-scale benchmark dataset for evaluating neural PDE solvers, featuring over 10K high-fidelity simulations of fluid flow and thermal interactions across diverse 2D/3D complex geometries, coupled with evaluation metrics and baseline results for machine learning models.

**Strengths:**

See above.

**Audience:**

Yes

**Claims And Evidence:**

NA

**Datasets And Benchmarks:**

NA

**Extended Submissions:**

NA

**Limitations:**

See above.

**Requested Changes:**

See above.

**Strengths And Weaknesses:**

Strengths:
- The paper is well-organized, with logical flow from problem motivation to dataset details, validation, and experiments.
- Figures and tables effectively illustrate dataset composition, geometry examples, and model performance, making technical content easier to grasp.

Weaknesses:
- In Section 3.1, the SBM formulation for Navier-Stokes (Equations 1-3) omits intermediate steps. How were the consistency, adjoint consistency, and penalty terms derived?
- For G3 (non-parametric shapes), Gaussian blur is applied to SkelNetOn data. How was the blur scale (σ=2) chosen?
- Section 4.2 states 2D LDC cases use a uniform 512×512 mesh, but Appendix A.1 mentions octree meshes. Are these meshes truly uniform, or do adaptive refinements occur near boundaries?
- How is the PDE residual computed numerically (e.g., finite differences, automatic differentiation)? For non-smooth solutions, could this metric introduce discretization errors?
- In Section 3.1, variables like w_i and u_i are introduced without explicit definitions. Are these standard notations in SBM literature, or do they require clarification?
- For FPO cases, 240 snapshots are provided. Why was the time step Δt=0.05 chosen? Does this resolution adequately capture high-frequency vortex shedding in transient flows?

---

### Review · Reviewer_yvNo · 2025-02-22

**Recommendation:** 3
**Confidence:** 2

**Summary Of Contributions:**

The paper introduces FlowBench, a large-scale benchmark dataset designed for evaluating machine-learning-based (or “neural”) PDE solvers on flow simulations over complex geometries. FlowBench is unique in that it:
1. Covers both 2D and 3D flow problems at multiple resolutions, including geometry-parameterized shapes and real-world-inspired shapes.
2. Incorporates multiphysics phenomena by including both Navier–Stokes and heat-transfer coupled simulations, at varying Reynolds and Grashof numbers (Re, Gr).
3. Provides high-fidelity “ground-truth” simulation results, including velocity, pressure, and temperature fields at multiple resolutions, plus derived engineering quantities (lift, drag, Nusselt number).
4. Encompasses 10K+ samples of fully resolved direct numerical simulations (DNS), making it significantly larger than existing open-source flow datasets.
5. Outlines evaluation metrics and baseline neural operator results (FNO, DeepONet, CNO, etc.) to facilitate systematic benchmarking of model accuracy, efficiency, and generalizability.

Taken together, FlowBench aims to stimulate progress on geometry-aware, multiphysics-capable PDE solvers in the machine learning community.

**Strengths:**

1. FlowBench addresses a gap for large-scale, geometry-rich flow simulation data with multiphysics. The dataset can serve as a “universal” test bed for neural PDE solvers and operator learning.
2. The dataset stands to benefit not just fluid dynamics specialists but also the AI-for-science community at large, who develop PDE surrogates or operator learners.
3. The authors carefully validated their HPC-based simulations. They also systematically included relevant dimensionless parameters (Re, Gr, Ri). The code references appear well documented.
4. The paper provides strong motivation, structured breakdown of geometry sets, boundary conditions, simulation parameters, and recommended evaluation metrics. Visual figures help convey essential details.

**Audience:**

Yes

**Broader Impact Concerns:**

If advanced PDE surrogates are created for aerodynamic or fluid dynamic optimization, they could also be used in military or surveillance contexts. However, the paper does not specifically address or foresee negative outcomes.

**Claims And Evidence:**

The main claims are that FlowBench is:
1. Larger (10K+ samples) than existing open-sourced flow dataset repositories,
2. Richer in geometry scope, including parametric, spherical-harmonic-based, and non-parametric shapes,
3. Providing multi-physics flow data (Navier–Stokes + heat transfer) with steady and unsteady regimes, and
4. Effective for training and benchmarking PDE surrogates, as validated by the authors’ baseline experiments.

These are well supported by the thorough data documentation (dimensions, shape generation, HPC simulation details), the validated references to similar prior benchmarks, and the authors’ demonstration of multiple neural operator baselines.

**Datasets And Benchmarks:**

The dataset is described in detail (geometry parameter generation, numerical validation, domain splits, multi-resolution fields). Besides, the authors mention it is hosted on Hugging Face (“BGLab/FlowBench”) with references to code. They also discuss specific usage instructions, recommended metrics, reference neural PDE solvers, and how to replicate baseline results.
As for the ethical and responsible use, no direct concerns regarding personal data or sensitive information arise, as it’s purely HPC-simulated fluid data. The authors do not mention specific licensing terms, but presumably it is open-access for research usage.

**Extended Submissions:**

The manuscript appears to present original dataset creation and does not read as an extended version of a prior publicly published paper.

**Limitations:**

1. The dataset focuses primarily on moderate Re (up to ~1000) and does not address strongly turbulent or compressible flows.
2. The shapes are 2D or 3D solid objects; hence flows with morphing or deforming boundaries are not included.
3. While the dataset is large, training neural PDE solvers to full resolution 3D fields will demand substantial GPU memory and extended training time, which might limit immediate adoption for researchers with limited compute.

**Requested Changes:**

Below are suggestions that would further strengthen the submission (none appear critically blocking, but all would be helpful):
1. If feasible, expand some subset of simulations to Re > 10^4 for evaluating truly turbulent flows.
2. Even just a small add-on dataset might help users who work with compressible or multiphase PDEs.
3. Provide an end-to-end tutorial or example of training advanced neural PDE solvers (like U-Net-based surrogates or advanced transformer-based operator networks) using FlowBench.
4. For the extremely large 3D fields, it might help if the authors share recommended chunking or loading strategies to handle memory constraints.

**Strengths And Weaknesses:**

### Strengths:
1. Presents large-scale dataset (~10,000 simulations), going beyond typical smaller PDE benchmarks.
2. Covers diverse geometries including parametric shapes, “blob-like” shapes via spherical harmonics, and non-parametric shapes (SkelNetOn-based).
3. Includes coupled flow and thermal data (Navier–Stokes + heat transfer) with varied boundary conditions.
4. Thorough baseline experiments with neural PDE solvers, plus recommended metrics for fair comparisons.

### Weakness:
1. The shapes in some categories (like skeleton-inspired geometry) are relatively random or abstract, which might not always match real-world industrial use cases.
2. Although recommended, the paper doesn’t delve deeply into domain shift scenarios involving fluid compressibility or advanced multiphase flows.

---

### Review · Reviewer_dUGs · 2025-03-01

**Recommendation:** 3
**Confidence:** 1

**Summary Of Contributions:**

This paper presents FlowBench — a large-scale benchmark design for evaluating neural PDE solvers for multiphysics flow simulation over complex geometrics. Includes various sets of geometrics with high-fidelity multiphysics flow simulation data of their velocity, pressure, and temperature.

Traditional PDE solvers are known for the high resource cost of traditional simulation approaches. In this paper, the authors offers a dataset at a much larger scale (>10x) and variety in terms of geometry and multiphysics compared to existing flow simulation datasets.

The authors use the Shifted Boundary Method (SMB) as the mathematical formulation to simulate and create the FlowBench, and then the dataset is validated by comparing them against the Boundary-Fitted Method. FlowBench enables the ML community to build, evaluate, and optimize ML-based PDE flow simulation solvers.

This paper also included sufficient dataset formatting information in section 4.5

Finally, the authors train and evaluate several neural PDE solvers on FlowBench by training them and measuring relative errors.

**Strengths:**

As detailed in the "Strengths" section above.

**Audience:**

Yes

**Claims And Evidence:**

Yes.

**Datasets And Benchmarks:**

All details are sufficient to me, except I do not see the maintenance plan.

**Extended Submissions:**

N/A

**Limitations:**

As detailed in the "Weakness" section above.

**Requested Changes:**

+ More detailed analysis to cover all proposed datasets in addition to 2D LDC-NS
+ Discussion to interpret the results in table 5-8
+ Data hosting and maintenance plan and code for evaluation as required by the https://data.mlr.press/acceptance-criteria.html

Generally, I feel this work is valuable for flow simulation and thus, addressing the above comments will likely secure my recommendation for acceptance. but given that I am not an expert in flow simulation thus I am also open to clarifications from the author if the author does not agree with some of the above-proposed changes.

**Strengths And Weaknesses:**

Strengths:

+ Large scale (>10x than existing datasets)
+ Modeling both complex geometries with multiphysics coverage
+ capturing a diverse range of physical phenomena, including Navier-Stokes and thermally coupled flows.
+ addresses a spectrum of canonical problems—such as flow past obstacles and lid-driven cavity flow with internal objects—spanning both two-dimensional and three-dimensional domains.
+ The paper is well structured and the writing is clear to me.

Weaknesses:

The evaluation part looks a little bit thin to me for:
+ It only covers the result of 2D LDC-NS which I believe would be better to cover in this paper rather than expecting community contribution.
+ Apart from listing evaluation errors in table 5-8, more discussion and deeper analysis on the interaction between each neural PDE solver and different geometry/dataset should be helpful to help the reader have a deeper understanding on the implications of each geometry/model.